# Constrained Best Arm Identification

**Tyron Lardy**
CWI and Leiden University

**Christina Katsimerou**
Booking.com
christina.katsimerou@booking.com

**Wouter M. Koolen**
CWI and University of Twente
wmkoolen@cwi.nl

## Abstract

In real-world decision-making problems, one needs to pick among multiple policies the one that performs best while respecting economic constraints. This motivates the problem of constrained best-arm identification for bandit problems where every arm is a joint distribution of reward and cost. We investigate the general case where reward and cost are dependent. The goal is to accurately identify the arm with the highest mean reward among all arms whose mean cost is below a given threshold. We prove information-theoretic lower bounds on the sample complexity for three models: Gaussian with fixed covariance, Gaussian with unknown covariance, and non-parametric distributions of rectangular support. We propose a combination of a sampling and a stopping rule that correctly identifies the constrained best arm and matches the optimal sample complexities for each of the three models. Simulations demonstrate the performance of our algorithms.

## 1 Introduction

In real-world decision-making systems, identifying the best policy is rarely a matter of optimizing a single performance metric in isolation. Effective policies often involve inherent trade-offs: they achieve desirable outcomes but incur costs. For instance, online platforms routinely deploy promotions to drive customer engagement [Zhao and Harinen, 2019, Zhang et al., 2024]. To ensure sustainability, such incentives must not only be effective but also economically viable—typically captured through feasibility constraints on budget or return on investment (ROI) [Goldenberg et al., 2020]. Similarly, bidding policies in online advertising need to drive traffic or purchases at an acceptable incremental ROI [Chen and Au, 2022]. In healthcare, a treatment needs to improve health outcomes under safety constraints. In all these examples, the goal is not simply to maximize the average benefit, but to do so subject to feasibility constraints—economic, operational, or ethical.

This motivates a constrained exploration problem: given a finite set of policies (arms), confidently identify the best policy with respect to a quality metric, among those that satisfy a feasibility constraint on a cost metric, such as a minimum return on investment (ROI), a risk threshold, or a fairness requirement. That is, each arm is associated with an unknown joint distribution on reward and cost, which may be arbitrarily dependent. During the exploration, the decision-maker chooses an arm in each round and observes a real-valued reward and cost, sampled from that arm. Once there is enough evidence that one arm has the highest quality amongst all feasible arms, or that no arm is feasible, the exploration stops and returns the best feasible arm or no arm respectively. This setting generalizes the classical best arm identification (BAI) problem in multi-armed bandits, where arms are judged only by their quality [Garivier and Kaufmann, 2016], and introduces an additional challenge not captured by standard BAI frameworks: coupled reward and cost metrics: business KPIs are rarely independent—more aggressive policies often yield higher rewards but also incur

greater costs. This interdependence violates the independence assumptions common in prior work on constrained bandits. In addition, reward and cost distributions are typically not parametric, especially in monetary applications.

In this paper, we provide a general strategy for the constrained BAI problem in the fixed-confidence setting. That is, we consider algorithms that are $\delta$-*correct*, i.e., that output the correct answer (either the best feasible arm or no arm) with probability at least $1 - \delta$ for some fixed confidence level $\delta$. The goal is to minimize the expected number of samples needed by the algorithm, while guaranteeing $\delta$-correctness.

Many variants of constrained BAI have previously been studied. Faizal and Nair [2022] and Yang et al. [2025] studied the same problem in the fixed budget setting, and Kone et al. [2025] in the Pareto set identification problem. An alternative way to incorporate costs in the BAI problem is the multi-fidelity formulation, in which known costs are tied to the desired accuracy level [Poiani et al., 2024], or by minimising the overall cost while still maximising a single reward dimension [Kanarios et al., 2024]. Wang et al. [2022] consider BAI with safety constraints with separate quality and feasibility dimensions. However, they assume independence and a linear or monotonic relation between the cost and reward. Furthermore, the constraints are required to be satisfied throughout the exploration, whereas we focus on pure exploration. David et al. [2018] and Hou et al. [2022] optimise a single performance metric, but with a constraint on some risk measure of the returned arm, such as its variance or a given quantile. Hu and Hu [2024] study a problem that is close to our setting. Each arm is associated with multiple performance metrics and their goal is to find the arm with the highest mean for a given metric, while a certain quantile of the others remains below a threshold. They assume only one of the metrics is sampled per rounds and that the different metrics are independent. Finally, Katz-Samuels and Scott [2019] propose a modification of the LUCB algorithm to solve the constrained BAI problem with multidimensional constraints. While they are the only one among the above to allow dependence, they focus purely on sub-Gaussian distributions, which is not always realistic for business purposes. Overall, most prior works ignore the dependence between reward and cost and/or focus on sub-Gaussian settings. To the best of our knowledge, none consider both dependence and arbitrary models, as we do here.

## 1.1 Contributions

Building on techniques from standard BAI [Garivier and Kaufmann, 2016, Degenne and Koolen, 2019], we derive instance-dependent lower bounds on the sample complexity for generic bandit models. The main difficulty therein lies in the fact that case distinctions arise from possible tradeoffs between cost and reward that do not occur when only considering the reward dimension. As is well-known in the BAI literature, the sample complexity lower bound also gives rise to the proportion of samples that an optimal algorithm should allocate to each arm. We show that these weights and the lower bound can be computed whenever we have numerical access to two transportation functions. In contrast to existing frameworks for BAI, which assume either exponential families [Garivier and Kaufmann, 2016] or nonparametric distributions [Agrawal et al., 2020], this allows us to treat all models in a unified framework.

We show that the transportation functions can be efficiently computed for three bivariate arm models: Gaussian with fixed $2 \times 2$ covariance matrix, Gaussian with unknown $2 \times 2$ covariance matrix, and non-parametric distributions of rectangular support. Our proposed algorithm then uses a plugin strategy to track these weights. As the stopping rule, we use a generalized-likelihood-ratio statistic, similar to e.g. Garivier and Kaufmann [2016], Degenne and Koolen [2019]. For proving $\delta$-correctness in the case of Gaussian with fixed covariance and non-parametric distributions of rectangular support, we import known results on the concentration for weighted sums of such statistics [Agrawal et al., 2021, Kaufmann and Koolen, 2021]. For the case of Gaussian with unknown covariance matrix, we prove a concentration result as proof of concept.

## 2 Sample complexity lower bounds

To set the stage, let $\mathcal{M}$ be a set of bivariate distributions on $\mathbb{R}^2$. For any $\nu \in \mathcal{M}$, we denote its mean reward and cost by $\boldsymbol{m}(\nu) = (m_1(\nu), m_2(\nu))$. We consider a $K$-armed bandit $\boldsymbol{\nu} = (\nu_1, \dots, \nu_K) \in \mathcal{M}^K$. At every time $n = 1, 2, \dots$, one arm $I_n \in [K]$ is chosen and a pair $X_n = (R_n, C_n)$ is drawn from $\nu_{I_n}$, where $R_n$ represents the obtained reward and $C_n$ the incurred cost. Given a

threshold $\gamma \in \mathbb{R}$, the objective is to identify the best feasible arm $i^*(\nu) = i^*(m(\nu))$, where $i^*(m) = \arg\max_{i:m_{i,2}\leq\gamma} m_{i,1}$. Here, the $\arg\max$ over the empty set is defined to be None, and we assume that the bandit $\nu$ has a unique best feasible arm $i^*(\nu) \in \mathcal{A} := [K] \cup \{\text{None}\}$. To avoid confusion, it will be crucial to distinguish between *arms* $[K]$ and *answers* $\mathcal{A}$, especially because for our problem these nearly coincide. Algorithm design must resolve the type conversion: which arm must be pulled to increase evidence in favor of a given answer?

Applying the generic lower bound of Garivier and Kaufmann [2016] results in the following lower bound on the sample complexity of any $\delta$-correct algorithm for constrained BAI (CBAI).

**Theorem 2.1** (Garivier and Kaufmann [2016])**.** *Let* $\delta \in (0,1)$. *For any $\delta$-correct strategy with stopping time $\tau_\delta$ and any bandit model $\nu \in \mathcal{M}^K$,*

$$\mathbb{E}_\nu[\tau_\delta] \geq T^*(\nu) \,\mathrm{KL}\,(\delta\|1-\delta),$$

*where* $\mathrm{KL}$ *denotes the Kullback-Leibler divergence and*

$$T^*(\nu)^{-1} = \max_{\boldsymbol{w}\in\triangle_K} \min_{\substack{\nu'\in\mathcal{M}^K \\ i^*(\nu)\neq i^*(\nu')}} \sum_{k=1}^{K} w_k \,\mathrm{KL}\,(\nu_k\|\nu'_k). \tag{1}$$

Their proof reveals that any strategy that matches the lower bound will also match $\boldsymbol{w}^*(\nu)$ as proportion of arm draws, where $\boldsymbol{w}^*(\nu)$ are the weights that achieve (1). To find a strategy with optimal sample complexity, we will compute the characteristic time $T^*(\nu)$ and corresponding oracle weights. To this end, we first present an abstraction, show how it still allows efficient computation, and then implement the abstraction for the following three models:

1. Gaussian with fixed covariance $\Sigma \succeq 0$: $\mathcal{M}_{G,\Sigma} := \left\{\mathcal{N}\,(\boldsymbol{\mu},\Sigma)\big|\boldsymbol{\mu}\in\mathbb{R}^2\right\}$.

2. Gaussian with unknown covariance: $\mathcal{M}_G := \left\{\mathcal{N}\,(\boldsymbol{\mu},\Sigma)\big|\boldsymbol{\mu}\in\mathbb{R}^2,\Sigma\succeq0\right\}$.

3. Non-parametric distributions on the unit square: $\mathcal{M}_B := \left\{P\big|P \text{ on } [0,1]^2\right\}$.

Other models can be worthwhile, for example modeling cost and reward as independent, each drawn from some single-parameter exponential family member. We focus on the above three models to highlight the role of dependent rewards and costs.

## 2.1 Solving (1) generically in terms of a transportation cost interface to the model

We analyze the CBAI characteristic time $T^*(\nu)$ from (1), and provide an efficient algorithm for computing $\boldsymbol{w}^*(\nu)$ and hence $T^*(\nu)$. To start, we introduce a shorthand for the KL projection of an arm $\nu \in \mathcal{M}$ onto the set of distributions $\{\nu' \in \mathcal{M} : m(\nu') = \boldsymbol{\mu}\}$ with a given mean $\boldsymbol{\mu} \in \mathbb{R}^2$:

$$\mathrm{KLinf}(\nu,\boldsymbol{\mu}) = \min_{\substack{\nu'\in\mathcal{M} \\ m(\nu')=\boldsymbol{\mu}}} \mathrm{KL}\,(\nu\|\nu'). \tag{2}$$

We suppress the dependence on $\mathcal{M}$ from the notation. The characteristic time (1) can be rewritten as

$$T^*(\nu)^{-1} = \max_{\boldsymbol{w}\in\triangle_K} \min_{\substack{\boldsymbol{\lambda}\in\mathbb{R}^{K\times2} \\ i^*(\nu)\neq i^*(\boldsymbol{\lambda})}} \sum_{k=1}^{K} w_k \,\mathrm{KLinf}(\nu_k,\boldsymbol{\lambda}_k). \tag{3}$$

At this point we see that we need to know about the mean vectors $\boldsymbol{\lambda}$ such that $i^*(\boldsymbol{\lambda}) \neq i^*$. We have

**Proposition 2.1.** *For each answer $i \in \mathcal{A}$, let* $\neg i := \left\{\boldsymbol{\lambda}\in\mathbb{R}^{K\times2}\big|i^*(\boldsymbol{\lambda})\neq i\right\}$. *Then*

$$\mathrm{cl}(\neg i) = \begin{cases} \bigcup_{j\neq i}\{\boldsymbol{\lambda}|\lambda_{j,1}\geq\lambda_{i,1} \text{ and } \lambda_{j,2}\leq\gamma\}\cup\{\boldsymbol{\lambda}|\lambda_{i,2}\geq\gamma\} & i \neq \text{None}, \\ \bigcup_{j\in[K]}\{\boldsymbol{\lambda}|\lambda_{j,2}\leq\gamma\} & i = \text{None}. \end{cases} \tag{4}$$

This shows in particular that the *Pure Exploration Rank* [Kaufmann and Koolen, 2021, Definition 20] of constrained BAI is *two*, as $\neg i$ is a union of parts in which at most two arms are constrained. This

will be useful in obtaining deviation thresholds below. For now, we use the partition above to simplify the characteristic time. Namely, if $i^*(\boldsymbol{\nu}) \neq \texttt{None}$, the problem of finding the characteristic time (3) reduces to

$$\max_{\boldsymbol{w} \in \triangle_K} \min \left\{ \min_{j \neq i^*} \left[ \min_{\substack{\boldsymbol{\lambda}_{i^*}, \boldsymbol{\lambda}_j \in \mathbb{R}^2: \\ \lambda_{i^*,1} \leq \lambda_{j,1} \\ \lambda_{j,2} \leq \gamma}} \sum_{k \in \{i^*, j\}} w_k \, \mathrm{KLinf}(\nu_k, \boldsymbol{\lambda}_k) \right], \min_{\substack{\boldsymbol{\lambda}_{i^*} \in \mathbb{R}^2: \\ \lambda_{i^*,2} \geq \gamma}} w_{i^*} \, \mathrm{KLinf}(\nu_{i^*}, \boldsymbol{\lambda}_{i^*}) \right\} \quad (5a)$$

and for $i^*(\boldsymbol{\nu}) = \texttt{None}$, it reduces to

$$\max_{\boldsymbol{w} \in \triangle_K} \min_j \min_{\substack{\boldsymbol{\lambda}_j \in \mathbb{R}^2: \\ \lambda_{j,2} \leq \gamma}} w_j \, \mathrm{KLinf}(\nu_j, \boldsymbol{\lambda}_j). \quad (5b)$$

We will argue that numerical access to the following two transportation cost functions, $c_1$ and $c_2$, suffices to implement this characteristic time $T^*$ and the corresponding oracle weights $\boldsymbol{w}^*$.

**Interface 2.1.** The following two functions need to be implemented efficiently:

1. the weighted cost for making arm $\nu_j$ beat arm $\nu_i$ (here $i$ can be assumed feasible)

$$c_1(\nu_i, \nu_j, w) := \min_{\substack{\boldsymbol{\lambda}_i, \boldsymbol{\lambda}_j \in \mathbb{R}^2 \\ \lambda_{i,1} \leq \lambda_{j,1} \text{ and } \lambda_{j,2} \leq \gamma}} \mathrm{KLinf}(\nu_i, \boldsymbol{\lambda}_i) + w \, \mathrm{KLinf}(\nu_j, \boldsymbol{\lambda}_j). \quad (6)$$

We need separate access to both terms of the sum at the minimum, that is, to $\mathrm{KLinf}(\nu_i, \boldsymbol{\lambda}_i^*)$ and $\mathrm{KLinf}(\nu_i, \boldsymbol{\lambda}_j^*)$. We will denote these by $c_{1,i}(\nu_i, \nu_j, w)$ and $c_{1,j}(\nu_i, \nu_j, w)$. We will also assume computational access to the limit $c_1(\nu_i, \nu_j, \infty) := \lim_{w \to \infty} c_1(\nu_i, \nu_j, w)$.

2. the cost for changing the feasibility status of an arm $\nu$

$$c_2(\nu) := \begin{cases} \min_{\substack{\boldsymbol{\lambda} \in \mathbb{R}^2 \\ \lambda_2 \geq \gamma}} \mathrm{KLinf}(\nu, \boldsymbol{\lambda}) & \text{if } m_2(\nu) \leq \gamma \\ \min_{\substack{\boldsymbol{\lambda} \in \mathbb{R}^2 \\ \lambda_2 \leq \gamma}} \mathrm{KLinf}(\nu, \boldsymbol{\lambda}) & \text{if } m_2(\nu) > \gamma \end{cases}.$$

In terms of Interface 2.1, our problem (5) simplifies to

$$T^*(\boldsymbol{\nu})^{-1} = \max_{\boldsymbol{w} \in \triangle_K} \begin{cases} \min \left\{ \min_{j \neq i^*} w_{i^*} c_1 \left( \nu_{i^*}, \nu_j, \frac{w_j}{w_{i^*}} \right), w_{i^*} c_2(\nu_{i^*}) \right\} & \text{if } i^* \neq \texttt{None}, \\ \min_{j \in [K]} w_j c_2(\nu_j) & \text{if } i^* = \texttt{None}. \end{cases} \quad (7)$$

To interpret the revealed structure, note that both cases minimize over precisely $K$ terms; one for each alternative answer different from the correct answer $i^*$. For the $i^* = \texttt{None}$ case, we find ourselves in a 1d thresholding problem Garivier et al. [2017], where the cost, $c_2$, is that of discriminating an arm from the threshold $\gamma$. For the $i^* \neq \texttt{None}$ case, the inner $\min_{j \neq i^*}$ sub-expression matches that of the transport cost for the best arm identification problem Garivier and Kaufmann [2016], where the cost to reverse the quality of two arms there is replaced by our $c_1$ (which in addition ensures feasibility of the second arm). The outer binary minimum adds one extra case to the range of possibilities to be considered, namely where the best looking arm is rendered infeasible.

We can solve the lower bound problem generically in terms of Interface 2.1:

**Theorem 2.2.** *Let $\boldsymbol{\nu}$ be a $K$-armed bivariate bandit. Let $i^* := i^*(\boldsymbol{\nu})$. For all $i \in [K]$, we have*

$$T^*(\boldsymbol{\nu}) = \begin{cases} \frac{\sum_{j=1}^K \tilde{w}_j(\tilde{C}^*)}{\tilde{C}^*} \\ \sum_{j=1}^K c_2(\nu_j)^{-1} \end{cases} \quad and \quad w_i^*(\boldsymbol{\nu}) = \begin{cases} \frac{\tilde{w}_i(\tilde{C}^*)}{\sum_{j=1}^K \tilde{w}_j(\tilde{C}^*)} & \text{if } i^* \neq \texttt{None} \\ \frac{c_2(\nu_i)^{-1}}{\sum_{j=1}^K c_2(\nu_j)^{-1}} & \text{if } i^* = \texttt{None} \end{cases}$$

*where $\tilde{w}_{i^*}(\tilde{C}) := 1$, and for each sub-optimal $j \neq i^*$, $\tilde{w}_j(\tilde{C})$ is the unique solution to $w$ in*

$$c_1(\nu_{i^*}, \nu_j, w) = \tilde{C}, \quad (8)$$

*and $\tilde{C}^*$ is the unique solution for $\tilde{C}$ in*

$$\sum_{j \neq i^*} \frac{c_{1,i^*}(\nu_{i^*}, \nu_j, \tilde{w}_j(\tilde{C}))}{c_{1,j}(\nu_{i^*}, \nu_j, \tilde{w}_j(\tilde{C}))} = 1 \quad (9)$$

*if it is attained in the interval $[0, \tilde{C}_{\max}]$ and $\tilde{C}^* = \tilde{C}_{\max}$ otherwise, where we abbreviate $\tilde{C}_{\max} := \min\{c_2(\nu_{i^*}), \min_{j \neq i^*} c_1(\nu_{i^*}, \nu_j, \infty)\}$.*

**Efficient Computation**   Note that this theorem is not only a characterisation, it unlocks a generic computational recipe given oracle access to $c_1$ and $c_2$. To see why, we first observe that the left-hand side of (8) is increasing in $w$, starting at 0 when $w = 0$, and reaching $c_1(\nu_{i^*}, \nu_j, \infty)$ for $w \to \infty$. Hence $w_j(\tilde{C})$ can be computed by binary search. Moreover, the proof reveals that the left-hand-side of (9) is increasing in $\tilde{C}$. So again, we can solve (9) for $\tilde{C}$ by binary search. All in all, we can compute $T^*$ and $\boldsymbol{w}^*$ using two nested binary searches. This is the same computational cost as the algorithm of [Garivier and Kaufmann, 2016, below Theorem 5] for the oracle weights in BAI. An efficient implementation for constrained BAI is the basis for the Track-and-Stop algorithm template. It therefore remains to implement $c_1$ and $c_2$ for each of our three arm models of interest.

## 2.2   Efficient Implementation of Interface 2.1 for our Three Models for Arms

We implement the interface functions $c_1$ and $c_2$ efficiently for our three arm models of interest. We also implement KLinf and discuss the effect of dependence.

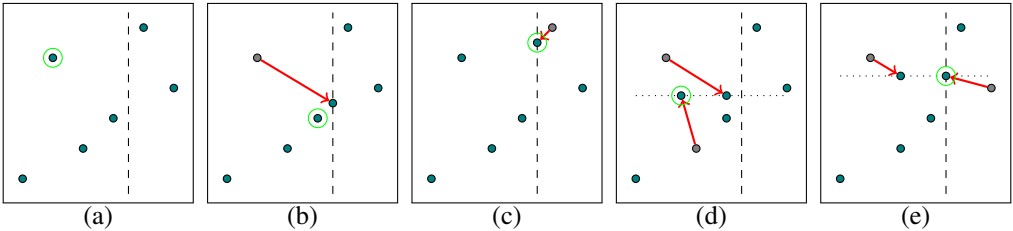

$$\hspace{3em}\text{(a)}\hspace{6em}\text{(b)}\hspace{6em}\text{(c)}\hspace{6em}\text{(d)}\hspace{6em}\text{(e)}$$

Figure 1: (a) shows the bivariate means of an example six-armed bandit $\boldsymbol{\nu}$, with its feasible best arm $i^*(\boldsymbol{\nu})$ circled green. The vertical axis is reward, while the horizontal axis is cost, with the vertical dashed line indicating the feasibility threshold $\gamma$. The four possible types of transports underlying (5a) are illustrated in (b)–(e). We can make the best arm infeasible (b). We can render an arm feasible that was already high (c). We can make a feasible arm exceed the best arm (d). And we can make an arm both feasible and better than the best arm (e). In these diagrams the reward-cost dependence within each arm manifests by the cheapest transports (indicated by red arrows) not being axis aligned.

### 2.2.1   Gaussian fixed covariance: KLinf, $c_1$ and $c_2$

**Proposition 2.2.** *Let* $\nu = \mathcal{N}(\boldsymbol{\mu}, \Sigma)$ *and consider* $\mathcal{M} = \mathcal{M}_{G,\Sigma}$. *Then*

$$\text{KLinf}(\nu, \boldsymbol{\lambda}) \;=\; \frac{1}{2} \, \|\boldsymbol{\mu} - \boldsymbol{\lambda}\|_{\Sigma^{-1}}^2 \qquad and \qquad c_2(\nu) \;=\; \frac{(\gamma - \mu_2)^2}{2\Sigma_{22}}. \tag{10}$$

*Moreover, let* $\nu_i = \mathcal{N}(\boldsymbol{\mu}_i, \Sigma)$ *and* $\nu_j = \mathcal{N}(\boldsymbol{\mu}_j, \Sigma)$ *with* $i^*(\{\boldsymbol{\mu}_i, \boldsymbol{\mu}_j\}) = i$, *then*

$$c_1(\nu_i, \nu_j, w) \;=\; \begin{cases} \frac{w(\mu_{j,2}-\gamma)^2}{2\Sigma_{22}} & \text{if } \mu_{j,1} - \frac{\Sigma_{12}}{\Sigma_{22}}(\mu_{j,2} - \gamma)_+ \geq \mu_{i,1} \\[1.2em] \frac{(\mu_{j,1}-\mu_{i^*,1})^2}{2\Sigma_{11}(1+\frac{1}{w})} & \text{if } \mu_{j,2} + \frac{\frac{1}{w}\Sigma_{12}}{\Sigma_{i,11}+\frac{1}{w}\Sigma_{11}}(\mu_{i,1}-\mu_{j,1}) \;\leq\; \gamma \\[1.2em] \dfrac{w\Sigma_{11}(\gamma-\mu_{j,2})^2 + |\Sigma| \left\| \begin{matrix} \mu_{j,1}-\mu_{i^*,1} \\ \mu_{j,2}-\gamma \end{matrix} \right\|_{\Sigma^{-1}}^2}{2\left(\Sigma_{11}\Sigma_{22}+|\Sigma|\frac{1}{w}\right)} & else. \end{cases}$$

Here, the case distinction in $c_1$ arises by first solving the infimum in (6) while forgetting about one of the constraints at a time. If the resulting minimizer happens to satisfy both constraints, then it must be the solution to the original problem, since its value is at least as low as that of the original problem. If this does not happen for either constraint, both of them must be active.

**The Impact of Dependence on Transportation Cost**   We are interested in the effect of dependence between reward and cost in all three models. The explicit formulas above allow us to highlight its effect explicitly. Here, dependence manifests as a nonzero covariance $\Sigma_{12} \neq 0$. To see its effect, we observe that the minimum cost to move an arm from mean $\boldsymbol{\mu}$ to a new location $\boldsymbol{\lambda}$ such that $\lambda_2 = \gamma$ is $c_2(\mathcal{N}(\boldsymbol{\mu}, \Sigma)) = \min_{\boldsymbol{\lambda}\in\mathbb{R}^2:\lambda_2=\gamma} \frac{1}{2}\|\boldsymbol{\mu} - \boldsymbol{\lambda}\|_{\Sigma^{-1}}^2 = \frac{(\gamma-\mu_2)^2}{2\Sigma_{22}}$, which is attained at $\boldsymbol{\lambda}^* = \left(\mu_1 + \frac{\Sigma_{12}}{\Sigma_{22}}(\gamma - \mu_2), \gamma\right)$. So we see that $\Sigma_{12} \neq 0$ causes the arm to move diagonally, even

though the objective was to move in dimension two. This is illustrated in Figure 1(b). On the other hand, counter intuitively, the *cost* of that move does not depend on $\Sigma_{12}$. These diagonal motions make it subtle to determine the active constraints for the $c_1$ motion (where we ask for a certain arm to be made feasible and better than another arm). E.g. an arm that starts feasible may be rendered infeasible by making it better. As visualized in Figure 1, the active constraints at the optimal solution can either be feasibility only (c), flipped mean reward only (d) or both (e).

### 2.2.2 Gaussian unknown covariance: KLinf, $c_1$ and $c_2$

For unknown covariance a subtlety arises: as we can see below, $\mathrm{KLinf}(\nu, \boldsymbol{\lambda})$ is not a convex function of $\boldsymbol{\lambda}$. As a result, $c_1$ as specified by (6) is not a convex optimization problem. Fortunately, $c_1$ can still be computed efficiently.

**Proposition 2.3.** *Let $\nu = \mathcal{N}(\boldsymbol{\mu}, \Sigma)$ and consider $\mathcal{M} = \mathcal{M}_G$. Then*

$$\mathrm{KLinf}(\nu, \boldsymbol{\lambda}) \;=\; \frac{1}{2} \ln\left(1 + \|\boldsymbol{\mu} - \boldsymbol{\lambda}\|^2_{\Sigma^{-1}}\right) \qquad \text{and} \qquad c_2(\nu) \;=\; \frac{1}{2} \ln\left(1 + \frac{(\gamma - \mu_2)^2}{\Sigma_{22}}\right). \quad (11)$$

*Moreover, let $\nu_i = \mathcal{N}(\boldsymbol{\mu}_i, \Sigma_i)$ and $\nu_j = \mathcal{N}(\boldsymbol{\mu}_j, \Sigma_j)$, then, abbreviating $\ell(x) := \frac{1}{2}\ln(1+x)$,*

$$c_1(\nu_i, \nu_j, w) \;=\; \min_{\theta \in \mathbb{R}} \; \ell\left(\frac{(\mu_{i,1} - \theta)^2_+}{\Sigma_{i,11}}\right) + w \begin{cases} 0 & \text{if } \mu_{i,2} \leq \gamma \text{ and } \mu_{j,1} \geq \theta \\ \ell\left(\frac{(\mu_{j,2} - \gamma)^2_+}{\Sigma_{j,22}}\right) & \text{if } \mu_{j,1} - \frac{\Sigma_{j,12}}{\Sigma_{j,22}}(\mu_{j,2} - \gamma)_+ \geq \theta \\ \ell\left(\frac{(\mu_{j,1} - \theta)^2_-}{\Sigma_{j,11}}\right) & \text{if } \mu_{j,2} + \frac{\Sigma_{j,12}}{\Sigma_{j,11}}(\mu_{j,1} - \theta)_- \leq \gamma \\ \mathrm{KLinf}(\nu_j, (\theta, \gamma)) & \text{else.} \end{cases}$$

Notice that the $c_2$ cost is fully determined by characteristics of the second dimension. In particular, it is independent of the dependence $\Sigma_{12}$ between the cost and reward dimension. This happens because the optimal move will take the covariance structure into account, which cancels its effect. Furthermore, the variable $\theta$ that appears in $c_1$ is introduced as a parameter such that $\lambda_{a,1} \leq \theta \leq \lambda_{a,2}$. With this extra parameter, the searches over $\lambda_{a,1}$ and $\lambda_{a,2}$ are straightforward. For the remaining search over $\theta \in \mathbb{R}$, it is possible to identify the points at which the active case in the second term switches; this will be instance dependent. For example, if $\mu_{i,2} \leq \gamma$ and $\Sigma_{i,12} < 0$, the second term will always be active, while for $\Sigma_{i,12} > 0$, case 3 takes over whenever $\theta \geq \mu_{i,1} + \Sigma_{i,11}\Sigma_{i,12}^{-1}(\gamma - \mu_{i,2})$. The optimal value on each segment can be found by setting the derivative of the objective to zero, which is a matter of finding the roots of a cubic. The global minimizer can then efficiently be found by comparing the minimizers on each segment. So computing $c_1$ or $c_2$ takes a constant amount of work.

### 2.2.3 Non-parametric supported on $[0,1]^2$: KLinf, $c_1$ and $c_2$

**Proposition 2.4.** *Let $\nu$ be a distribution on $[0,1]^2$ and consider $\mathcal{M} = \mathcal{M}_B$. Furthermore, let $\mathcal{R}_{\boldsymbol{\lambda}} := \{(a_1, a_2) \in \mathbb{R}^2 \mid \forall x \in [0,1]^2 : 1 + a_1(x_1 - \lambda_1) + a_2(x_2 - \lambda_2) \geq 0\}$. Then*

$$\mathrm{KLinf}(\nu, \boldsymbol{\lambda}) \;=\; \max_{a_1, a_2 \in \mathcal{R}_{\boldsymbol{\lambda}}} \; \mathop{\mathbb{E}}_{X \sim \nu} \left[\ln\left(1 + a_1(X_1 - \lambda_1) + a_2(X_2 - \lambda_2)\right)\right], \quad (12)$$

$$c_2(\nu) \;=\; \max_{a \in [\frac{-1}{1-\gamma}, \frac{1}{\gamma}]} \; \mathop{\mathbb{E}}_{X \sim \nu} \left[\ln\left(1 + a(X_2 - \gamma)\right)\right].$$

*Finally, let $\nu_i, \nu_j$ be distributions on $[0,1]^2$. With $\mathcal{R}'_w := \{\boldsymbol{b} \in \mathbb{R}^3 \mid b_3 \geq 0 \geq b_2, \forall x \in [0,1]^2 : 1 - w(b_1 + b_2 x_1) \geq 0 \text{ and } 1 + b_1 + b_2 x_1 + b_3(x_2 - \gamma) \geq 0\}$, we have*

$$c_1(\nu_i, \nu_j, w) \;=\; \max_{\boldsymbol{b} \in \mathcal{R}'_w} \; \mathop{\mathbb{E}}_{\nu_i}\left[\ln(1 - w(b_1 + b_2 X_1))\right] + w \mathop{\mathbb{E}}_{\nu_j}\left[\ln(1 + b_1 + b_2 X_1 + b_3(X_2 - \gamma))\right].$$

Note that when $\boldsymbol{\lambda} \in (0,1)^2$, the region $\mathcal{R}_{\boldsymbol{\lambda}}$ is a compact convex set in $\mathbb{R}^2$. In fact, being the intersection of four half-spaces, it is a quadrilateral with its four vertices on the axes. Moreover, the objective is concave in $\boldsymbol{\lambda}$. This means that for $\nu$ a distribution of finite support (e.g. an empirical distribution) we can compute $\mathrm{KLinf}(\nu, \boldsymbol{\lambda})$ using off-the-shelf convex optimisation methods e.g. the ellipsoid method. Similarly, $\mathcal{R}'_w$ for $w > 0$ is a compact convex subset of $\mathbb{R}^3$, being an intersection of six half spaces.

The case distinction that we saw for $c_1$ in the Gaussian fixed covariance case did not disappear. In fact, it manifests in the region for $\boldsymbol{b}$: $b_3$ is the Lagrange multiplier for enforcing feasibility of arm $j$, and $-b_2$ is the Lagrange multiplier for enforcing the correct order of mean rewards. Either (but not both) can be zero at optimality if the respective constraint is satisfied already.

## 3 Asymptotically Optimal Algorithm

We now develop an asymptotically optimal algorithm. As this is classic, we defer details to the appendix. Throughout, we denote by $N_i(n) = \sum_{s=1}^{n} \mathbf{1}_{I_s=i}$ the number of samples taken from arm $i$ in the first $n$ rounds.

**Estimates** Our approach is based on an estimate $\hat{\boldsymbol{\nu}}(n)$ of the bandit $\boldsymbol{\nu}$ after $n$ samples. For the fixed covariance case, we let $\hat{\nu}_i(n) := \mathcal{N}(\hat{\mu}_i(n), \Sigma)$, where $\hat{\mu}_i(n) = \frac{1}{N_i(n)} \sum_{s=1}^{n} \mathbf{1}_{I_s=i} X_s$ is the empirical mean of arm $i$ after $n$ bivariate outcomes $X_s = (R_s, C_s)$. For the unknown covariance case, we let $\hat{\nu}_i(n) := \mathcal{N}(\hat{\mu}_i(n), \hat{\Sigma}_i(n))$, where $\hat{\Sigma}_i(n)$ is the empirical covariance of the samples from arm $i$. Finally, for the non-parametric case, we let $\hat{\nu}_i(n)$ be the empirical distribution of the samples $\{X_s \mid s \leq n, I_s = i\}$ from arm $i$.

**Stopping and recommendation rule** The Generalised Likelihood Ratio (GLR) statistic is defined, for $\hat{\imath} = i^*(\hat{\boldsymbol{\nu}}(n))$, by

$$\Lambda_n := \begin{cases} \min\left\{ \min_{j \neq \hat{\imath}} N_{\hat{\imath}}(n) c_1\left(\hat{\nu}_{\hat{\imath}}(n), \hat{\nu}_j(n), \frac{N_j(n)}{N_{\hat{\imath}}(n)}\right), N_{\hat{\imath}}(n) c_2(\hat{\nu}_{\hat{\imath}}(n)) \right\} & \text{if } \hat{\imath} \neq \texttt{None}, \\ \min_{j \in [K]} N_j(n) c_2(\hat{\nu}_j(n)) & \text{if } \hat{\imath} = \texttt{None}. \end{cases} \quad (13)$$

We stop at the first time $\tau_\delta := \inf\{n \in \mathbb{N} : \Lambda_n \geq \beta(\delta, n)\}$ the GLR crosses the exploration threshold $\beta$ given below, and at that point we will recommend the empirical best feasible arm $\hat{\imath} := i^*(\hat{\boldsymbol{\nu}}(n))$. We show in Appendix B that

**Theorem 3.1.** *The following exploration thresholds result in a $\delta$-correct recommendation*

$$\beta_{G,\Sigma}(\delta, n) = \ln \frac{K}{\delta} + 4 \ln \frac{\ln \frac{K}{\delta}}{4} + 8 \ln(4 + \ln n/2) \qquad \text{once } \ln \frac{1}{\delta} \geq 6,$$

$$\beta_G(\delta, n) = \ln \frac{K}{\delta} + 2 \ln n + 4 \ln \ln n + 2 \ln \left( \ln \frac{K}{\delta} + 2 \ln n + 4 \ln \ln n \right),$$

$$\beta_B(\delta, n) = \ln \frac{K}{\delta} + 2 + 4 \ln(1 + n/2).$$

These thresholds all account for confidence ($\ln \frac{1}{\delta}$), a union bound across incorrect answers ($\ln K$), and a courser ($\ln n$) or finer ($\ln \ln n$) union bound across time. These bounds are conservative in practice; in the experiments we will use $\ln \frac{1}{\delta} + \ln \ln n$ instead (and verify that the rate of incorrect recommendations remains below $\delta$).

**TaS** Finally, our sampling rule ensures the asymptotic optimality. We compute a plug-in estimate of the oracle weights $\boldsymbol{w}_n := \boldsymbol{w}^*(\hat{\boldsymbol{\nu}}(n))$, and pick $I_n = \arg\min_i N_i(n-1) - \sum_{s=1}^{n-1} w_{s,i}$ (C-Tracking). We add forced exploration to keep $N_i(n) \geq \sqrt{n}$. All in all, these ingredients guarantee

**Theorem 3.2.** *TaS is asymptotically optimal, i.e.* $\lim_{\delta \to 0} \frac{\mathbb{E}_{\boldsymbol{\nu}}[\tau_\delta]}{\ln \frac{1}{\delta}} = T^*(\boldsymbol{\nu})$.

## 4 Simulations

In this section, we put our algorithm, **TaS**, to the test on the four bandits depicted in Figure 2. We treat both the unknown-covariance Gaussian model and the bounded model. For the latter, we clip the Gaussian arms from Figure 2 to $[0, 1]^2$. Since there are no off-the-shelf algorithms to compare to, we adjust a number of sampling strategies used in the BAI literature to the constrained BAI setting.

For the Gaussian model with unknown covariance, the **EV-TaS** sampling rule uses the empirical covariance to track the weights as if we were in the fixed covariance model. A similar rule was

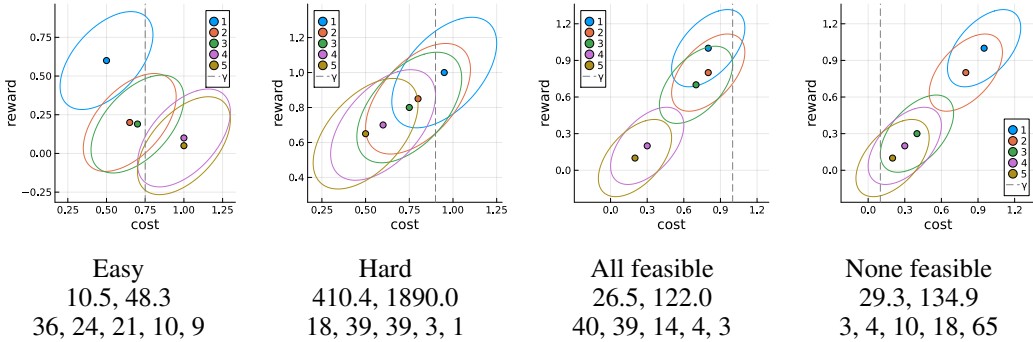

| Easy | Hard | All feasible | None feasible |
| 10.5, 48.3 | 410.4, 1890.0 | 26.5, 122.0 | 29.3, 134.9 |
| 36, 24, 21, 10, 9 | 18, 39, 39, 3, 1 | 40, 39, 14, 4, 3 | 3, 4, 10, 18, 65 |

Figure 2: Each diagram illustrates a 5-arm bandit $\boldsymbol{\nu}$ with Gaussian arms. The dots and ellipses give the mean and one standard deviation ring around it in covariance matrix $\Sigma = [0.1\ 0.05; 0.05\ 0.09]$. The two numbers and vector are $T^*(\boldsymbol{\nu})$, $T^*(\boldsymbol{\nu}) \ln \frac{1}{\delta}$ for $\delta = 0.01$, and $\boldsymbol{w}^*(\boldsymbol{\nu})$ in percentages.

previously considered for regular BAI by Jourdan et al. [2023]. Note that there is no reason to believe that this should work well (and it does not), as the sampling proportions will be sub-optimal. However, it is reasonable to consider if one does not know how to properly handle the unknown covariance. For the bounded case, we introduce the **GA-TaS** [Ménard, 2019], instead of the original TaS, due to the high cost of computing the optimal sampling ratios per round. Instead we use gradient ascent to solve the optimization problem online, and thus more efficiently.

**TopTwo-TCI** sampling rule is based on Top Two algorithms in regular BAI, where the arm to sample at each time is randomly chosen between a leader and challenger [Jourdan et al., 2022]. We thus need to define what the leader and challenger mean in our case. The current best answer $\hat{i}_n = i^*(\hat{\boldsymbol{\nu}}(n))$ will serve as the leader. For the challenger, note that the GLR ($\Lambda_n$ as in (13)) is defined as a minimum of $K$ terms, each corresponding to an answer different from $\hat{i}_n$. Let us denote each of these terms by $\Lambda_{n,j}$. As challenger, we take the answer that minimizes $\Lambda_{n,j} + \log(N_j(n))$, where we let $N_{\texttt{None}}(n) := N_{\hat{i}_n}(n)$. If either the challenger or the leader is None, we sample the other deterministically. This setup resembles the best challenger implementation of Hu and Hu [2024], with the substantial difference that our implementation regards a constrained mean rather than a quantile, and our selection criterion for the challenger accounts for the dependence between reward and cost dimensions. **Oracle** samples all arms with the optimal weights for the true model. **Racing** repeatedly samples uniformly all arms and eliminates an answer $j$ (and the corresponding arm if $j \neq \texttt{None}$) once it can be rejected. That is, if $\hat{i} = \texttt{None}$, we eliminate $j$ when $\Lambda_{n,j} \geq \beta(n, \delta)$, i.e. feasibility of arm $j$ is implausible. If $\hat{i} \neq \texttt{None}$, we eliminate answer $j$ when $\min\{\Lambda_{n,j}, \Lambda_{n,\texttt{None}}\} \geq \beta(n, \delta)$. The second term ensures that it is implausible that $\hat{i}$ is infeasible. If $j \neq \texttt{None}$, the first term in addition ensures that it is implausible that $j$ is better than $\hat{i}$. We keep sampling until one answer remains. **Uniform** samples all arms in a round robin fashion. **TaS-1d** solves the unconstrained BAI problem, ignoring the cost dimension.

All algorithms use the same GLR rule and the stylized stopping threshold $\log(1/\delta) + \log\log(t)$, originally used by Garivier and Kaufmann [2016] and heavily adopted in the literature for allowing shorter runtimes while keeping the errors lower than $\delta$. As initialization, we start by pulling each arm 3 times, which is the minimum required for the covariance matrix estimation. We work in the moderate confidence regime of $\delta = 0.01$. All instances were repeated 1000 times, except the hard one, which we ran 500 times. The results are shown in Table 1 for the unknown covariance Gaussian case and Table 2 for the bounded case. All empirical error rates remain below $\delta$.

Table 1: Gaussian unknown covariance: average runtimes with standard errors

| Instance | TaS-EV | TaS | Oracle | Uniform | TopTwo-TCI | Racing | TaS-1d |
|---|---|---|---|---|---|---|---|
| Easy | 79.4± 0.7 | 76.3± 0.7 | 89.6± 0.4 | 136.4± 0.6 | 68.8± 0.6 | 136.7± 1.5 | 96.6± 0.5 |
| Hard | 3291.1± 70.0 | 3218.7± 68.6 | 4225.4± 59.4 | 5498.9±129.8 | 2859.9± 54.9 | 2864.6± 51.4 | 4815.9±101.6 |
| All feasible | 199 ± 2.5 | 190.4± 2.4 | 229.7± 2.3 | 354.0± 4.8 | 174.6± 2.4 | 271.0± 2.6 | 186.4± 2.3 |
| None feasible | 234.2± 2.7 | 222.6± 2.6 | 270.1± 3.1 | 576.0± 13.4 | 241.9± 5.4 | 174.6± 2.2 | 3293.4± 84.4 |

Table 2: Bounded (non-parametric on $[0, 1]^2$): average runtimes with standard errros

| Instance | GA-TaS | Oracle | Uniform | TopTwo-TCI | Racing |
|---|---|---|---|---|---|
| Easy | 98.8± 0.9 | 104.7± 0.9 | 120.7± 1.2 | 96.4± 0.8 | 114.5± 0.9 |
| Hard | 539.7± 7.6 | 669.3± 6.6 | 1088.4±13.0 | 457.3± 4.7 | 933.4±17.8 |
| All feasible | 241.0± 3.5 | 256.7± 2.8 | 409.8± 6.1 | 191.1± 2.9 | 221.7± 2.8 |
| None feasible | 69.6± 0.7 | 68.2± 0.7 | 116.9± 1.8 | 47.5± 0.5 | 49.0± 0.5 |

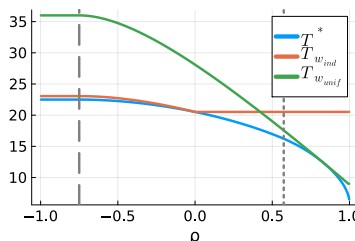 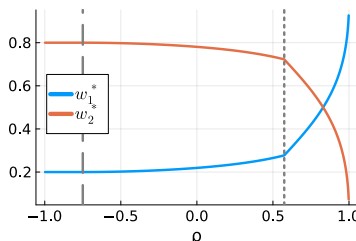

Figure 3: Sample complexity and optimal weights as a function of dependence $\rho$.

## 5 The Impact of Dependence on the Sample Complexity

In this section we study the impact of dependence on the sample complexity. We study the following two-arm problem $\nu_\rho$ in the fixed covariance Gaussian model as a function of correlation $\rho \in [-1, 1]$: the feasibility threshold is $\gamma = \frac{2}{3}$, the arm means are $\mu_1 = (0, 0)$ and $\mu_2 = \left(-\frac{1}{4}, 1\right)$, cost and reward each have variance $\Sigma_{11} = \Sigma_{22} = 1$, and the correlation between them is $\Sigma_{12} = \rho$. These parameters were selected to illustrate the three possible regimes we discuss below. Note that arm 1 is feasible while arm 2 is not, so the correct answer is always arm 1. Figure 3 shows the characteristic time $T^*(\nu_\rho)$ as a function of $\rho$. It is contrasted with the number of samples needed when sampling uniformly, $T_{w_{\text{unif}}}(\nu_\rho)$, and when using the optimal sample weights ignoring the dependence, $T_{w_{\text{ind}}}(\nu_\rho)$ (by assuming $\rho = 0$). That is, the sample complexity for when we sample according to one of these rules, but still stop with the 'correct' GLR rule. Any other $\delta$-correct stopping rule would be slower.

By inspecting (7), we see that the optimal unnormalized weight on the second arm will be chosen to maximize the cost of making arm two better than arm one, as long as that is cheaper than making arm one infeasible. This corresponds to the case that $\tilde{C}^*$ is attained in the interval $[0, \tilde{C}_{\max}]$ in Theorem 2.2. Notice that the cost of changing the feasibility status is independent of $\rho$, as can be seen in the expression for $c_2$ in Proposition 2.2. The mean reward of arm two is moved to $\mu_{2,1} - \rho(\mu_{2,2} - \gamma)$, which does depend on $\rho$. This will result in arm two becoming better than arm one for $\rho < \frac{\mu_{1,1} - \mu_{2,1}}{\gamma - \mu_{2,2}}$. Therefore, the cost of making arm two feasible and better equals the cost of just making it feasible; see also case 1 of $c_1$ in Proposition 2.2. This results in the flat regime on the left of the dashed line in Figure 3. As $\rho$ further increases, making arm two feasible and better will involve moving both arms, the cost of which does depend on $\rho$, as can be seen in the third case of $c_1$ in Proposition 2.2. At some point, the maximum of this cost over the unnormalized weight becomes larger than the cost of making arm one infeasible. The optimum unnormalized weight will then make the transportation cost equal to the cost of making arm one infeasible, corresponding to the case that $\tilde{C}^* = c_2(\nu_{i^*})$ in Theorem (2.2). This causes the change in behavior at the dotted line.

Finally, it is noteworthy that, in this case, uniform sampling sometimes outperforms the strategy assuming independence. This occurs because, for large values of $\rho$, uniform weights more closely resemble the optimal weights than those derived under the assumption of independence.

## 6 Discussion

We introduced the constrained best arm problem (CBAI), where each arm hides a joint distribution of reward and cost. The goal is to identify from observations the arm of highest mean reward among all arms with mean cost below a given threshold, or to report None if all arms are infeasible. This

model in particular allows us to study the impact of dependence between cost and reward. We characterized optimal sample complexity, and implemented the resulting optimal algorithms for three classes of arm distributions. We analyzed our algorithms theoretically, and showed that they are asymptotically optimal. Finally, we experimentally investigated the performance of a variety of algorithmic templates, including Track-And-Stop, Top-Two and Racing, and show that they perform well. We now discuss questions and future directions.

**Can one handle multiple constraints?** Let us imagine a multivariate problem with all-but-one dimensions being costs needing to be below respective thresholds. What would change? We could still have a $c_1$ and $c_2$ decomposition, where $c_1$ is the cost to make a designated arm feasible and better, while $c_2$ is the cost of toggling the feasibility status of an arm. For all our three models, changing the feasibility status of an arm is a solvable problem, even with multiple constraints. We either enforce all of them, or enumerate all constraints to violate, and optimize $\mathrm{KLinf}$. What gets tricky is making an arm feasible and better than another one. In the known $\Sigma$ and bounded cases, this still results in a convex problem. Yet in the unknown $\Sigma$ case, $\mathrm{KLinf}(\nu, \boldsymbol{\lambda})$ is not convex in $\boldsymbol{\lambda}$ (but it is quasi-convex) and $c_1$ is not even a quasi-convex problem in $\boldsymbol{\lambda}$ (as it is a sum of quasi-convex objectives). We currently optimize $c_1$ in Proposition 2.3 by locating the optimal $\theta$ by finding roots of a small number of cubics. With multiple constraints, the number of cases in the right-most function of $\theta$ in Proposition 2.3 may equal the (exponential) number of subsets of active constraints.

**Is the stylized threshold valid?** In Theorem 3.1, we propose GLR thresholds that are of the form $\ln \frac{1}{\delta}$ with an added correction factor of $\ln \ln n$ for the known covariance model and $\ln n$ for the unknown covariance and bounded models. In the simulations, we used a stylized threshold of $\ln \ln n$ for all models. For known covariance, $\ln \ln n$ is proven valid using mixture martingale techniques Kaufmann and Koolen [2021]. For Gaussians with unknown covariance, it is also possible to relate the GLR statistic to a mixture martingale, as noted in the one-dimensional case by Wang and Ramdas [2025]. However, using this to achieve a $\ln \ln n$ threshold would require a sophisticated argument about the exact nature of this relation, which we leave for future work.

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

# A  Proofs for Section 2

## A.1  Proof of Proposition 2.1

Technically, below we characterize the closure of $\neg i$. For the value of optimization problems constrained to $\neg i$ this difference is immaterial, yet it ensures the optimizers are attained.

First, suppose that $i = \texttt{None}$. Any $\boldsymbol{\lambda}$ with $i^*(\boldsymbol{\lambda}) \neq i$ must have an arm $j$ such that $\lambda_{j,2} \leq \gamma$. Conversely, any $\boldsymbol{\lambda}$ that has an arm $j$ with $\lambda_{j,2} \leq \gamma$ has $i^*(\boldsymbol{\lambda}) \neq i$. So we find that $\neg i = \cup_{j \in [K]} \{\boldsymbol{\lambda} | \lambda_{j,2} \leq \gamma\}$ for $i = \texttt{None}$.

For $i \neq \texttt{None}$, any $\boldsymbol{\lambda}$ with $i^*(\boldsymbol{\lambda}) \neq i$ must either have an arm $j$ that is both feasible and better than $i$, or arm $i$ must not be feasible. That is, either there exists $j$ such that $\lambda_{j,1} \geq \lambda_{i,1}$ and $\lambda_{j,2} \leq \gamma$ or $\lambda_{i,2} \geq \gamma$. Conversely, any $\boldsymbol{\lambda}$ for which either $\lambda_{j,1} \geq \lambda_{i,1}$ and $\lambda_{j,2} \leq \gamma$ or $\lambda_{i,2} \geq \gamma$ must necessarily have $i^*(\boldsymbol{\lambda}) \neq i$. Therefore, $\neg i = \cup_{j \neq i} \{\boldsymbol{\lambda} | \lambda_{j,1} \geq \lambda_{i,1} \text{ and } \lambda_{j,2} \leq \gamma\} \cup \{\boldsymbol{\lambda} | \lambda_{i,2} \geq \gamma\}$.

## A.2  Proof of Theorem 2.2

We first handle the case that $i^*(\boldsymbol{m}) \neq \texttt{None}$. We are interested, following (5a), in

$$\max_{\boldsymbol{w} \in \triangle_K} \min \left\{ \min_{j \neq i^*} w_{i^*} c_1 \left( \nu_{i^*}, \nu_j, \frac{w_j}{w_{i^*}} \right), w_{i^*} c_2(\nu_{i^*}) \right\}.$$

This can be restructured to

$$\max_{w_{i^*} \in [0,1]} \min \{ g(w_{i^*}), w_{i^*} c_2(\nu_{i^*}) \} \tag{14}$$

where

$$g(w_{i^*}) := \max_{\boldsymbol{w}_{-i^*} \in (1-w_{i^*})\triangle_{K-1}} \min_{j \neq i^*} w_{i^*} c_1 \left( \nu_{i^*}, \nu_j, \frac{w_j}{w_{i^*}} \right).$$

Now each $w_j$ for $j \neq i^*$ only appears in one term of the minimum. This means that the optimal solution for $\boldsymbol{w}_{-i^*}$ is to balance all contributions. We hence need to solve the system

$$C = w_{i^*} c_1 \left( \nu_{i^*}, \nu_j, \frac{w_j}{w_{i^*}} \right) \tag{15a}$$

$$1 - w_{i^*} = \sum_{j \neq i^*} w_j \tag{15b}$$

for $\boldsymbol{w}_{-i^*}$ and $C$, and then the value is $C$. Note that $C$ is a concave function of $w_{i^*}$. Resuming from (14), and introducing $\tilde{C} = C/w_{i^*}$ and $w_j(C, w_{i^*}) = w_{i^*} w_j(C/w_{i^*}, 1) = w_{i^*} \tilde{w}_j(\tilde{C})$, where $\tilde{w}_j(\tilde{C})$ is the solution for $\tilde{w}_j$ in

$$\tilde{C} = c_1 \left( \nu_{i^*}, \nu_j, \tilde{w}_j \right),$$

we are left with

$$\max_{\substack{C, w \\ \text{s.t. } (15)}} \min\{C, w_{i^*} c_2(\nu_{i^*})\} = \max_{\tilde{C}} \frac{\min\{\tilde{C}, c_2(\nu_{i^*})\}}{1 + \sum_{j \neq i^*} \tilde{w}_j(\tilde{C})}$$

where in particular we solved for $w_{i^*} = \frac{1}{1 + \sum_{j \neq i^*} \tilde{w}_j(\tilde{C})}$ and $w_j = \frac{\tilde{w}_j(\tilde{C})}{1 + \sum_{j \neq i^*} \tilde{w}_j(\tilde{C})}$. The objective is decreasing for $\tilde{C} \geq c_2(\nu_{i^*})$, so the maximum is between $0$ and that. To find it, we need to cancel (or, for bisection, compute the sign of) the derivative, i.e.

$$0 = \frac{1 + \sum_{j \neq i^*} \tilde{w}_j(\tilde{C}) - \tilde{C} \sum_{j \neq i^*} \tilde{w}_j'(\tilde{C})}{(1 + \sum_{j \neq i^*} \tilde{w}_j(\tilde{C}))^2}$$

and that is equivalent to

$$1 = \sum_{j \neq i^*} \left( \tilde{C} \tilde{w}_j'(\tilde{C}) - \tilde{w}_j(\tilde{C}) \right).$$

Differentiating the definition of $\tilde{w}_j(\tilde{C})$, we find

$$1 = c_1' \left( \nu_{i^*}, \nu_j, \tilde{w}_j(\tilde{C}) \right) \tilde{w}_j'(\tilde{C})$$

and it hence remains to solve for

$$1 = \sum_{j \neq i^*} \frac{c_1\left(\nu_{i^*}, \nu_j, \tilde{w}_j\right) - \tilde{w}_j(\tilde{C}) c_1'\left(\nu_{i^*}, \nu_j, \tilde{w}_j(\tilde{C})\right)}{c_1'\left(\nu_{i^*}, \nu_j, \tilde{w}_j(\tilde{C})\right)} = \sum_{j \neq i^*} \frac{\text{KLinf}(\nu_{i^*}, \boldsymbol{\lambda}_{i^*}(\tilde{C}))}{\text{KLinf}(\nu_j, \boldsymbol{\lambda}_j(\tilde{C}))}.$$

This is the same as equating $F(\tilde{C}) = 1$, where $F(\tilde{C}) := \sum_{j \neq i^*} \tilde{C} \tilde{w}_j'(\tilde{C}) - \tilde{w}_j(\tilde{C})$. For $\tilde{C}_1 > \tilde{C}_2$, we see

$$
\begin{aligned}
F(\tilde{C}_1) - F(\tilde{C}_2) &= \sum_{j \neq i^*} \tilde{C}_1 \tilde{w}_j'(\tilde{C}_1) - \tilde{C}_2 \tilde{w}_j'(\tilde{C}_2) + \tilde{w}_j(\tilde{C}_2) - \tilde{w}_j(\tilde{C}_1) \\
&\geq \sum_{j \neq i^*} \tilde{C}_1 \tilde{w}_j'(\tilde{C}_1) - \tilde{C}_2 \tilde{w}_j'(\tilde{C}_2) - \tilde{w}_j'(\tilde{C}_1)\left(\tilde{C}_2 - \tilde{C}_1\right) \\
&= \sum_{j \neq i^*} C_2(\tilde{w}_j'(C_1) - \tilde{w}_j'(C_2)) > 0,
\end{aligned}
$$

where we use a tangent bound on $\tilde{w}_j$ together with the fact that $\tilde{w}_j'$ is increasing, since $\tilde{w}_j$ is the inverse of a concave function. It follows that $F(\tilde{C})$ is increasing, so that $F(\tilde{C}) = 1$ can be found through bisection.

If $i^*(\boldsymbol{m}) = \texttt{None}$, then, following (5b), we are interested in computing

$$\max_{\boldsymbol{w} \in \triangle_K} \min_j w_j c_2(\nu_j).$$

By reasoning similar to the above, the optimal $\boldsymbol{w}$ will balance all contributions. It follows that for every arm $i$

$$w_i^* = \frac{c_2(\nu_i)^{-1}}{\sum_j c_2(\nu_j)^{-1}}.$$

### A.3 Proofs for the Gaussian transportation functions

In this section, we provide the proofs of Propositions 2.2 and 2.3. For both, we will need the KL between two Gaussians. That is, for $\nu = \mathcal{N}(\boldsymbol{\mu}, \Sigma)$ and $\nu' = \mathcal{N}(\boldsymbol{\lambda}, \Sigma')$, we have

$$\text{KL}(\nu\|\nu') = \frac{1}{2}\left(-\log\frac{|\Sigma|}{|\Sigma'|} - 2 + \text{tr}(\Sigma'^{-1}\Sigma) + (\boldsymbol{\mu} - \boldsymbol{\lambda})^T \Sigma'^{-1}(\boldsymbol{\mu} - \boldsymbol{\lambda})\right). \tag{16}$$

It furthermore helps to know that, for fixed $\lambda_1$, the minimum 2-dimensional KL is in fact the 1-dimensional KL (and this insight can symmetrically be applied with dimension 1 and 2 exchanged):

$$\min_{\lambda_2} (\boldsymbol{\mu} - \boldsymbol{\lambda})^T \Sigma'^{-1}(\boldsymbol{\mu} - \boldsymbol{\lambda}) = \frac{(\mu_1 - \lambda_1)^2}{\Sigma_{11}'}, \quad \text{achieved at} \quad \lambda_2 = \mu_2 - \frac{\Sigma_{12}'}{\Sigma_{11}'}(\mu_1 - \lambda_1). \tag{17}$$

We now proceed with the proofs.

#### A.3.1 Proof of proposition 2.2

For $\nu = \mathcal{N}(\boldsymbol{\mu}, \Sigma)$ and $\mathcal{M} = \mathcal{M}_{G,\Sigma}$ we have, by (16),

$$\text{KLinf}(\nu, \boldsymbol{\lambda}) = \text{KL}(\nu\|\nu') = \frac{1}{2}\|\boldsymbol{\mu} - \boldsymbol{\lambda}\|_{\Sigma^{-1}}^2,$$

where $\nu' = \mathcal{N}(\boldsymbol{\lambda}, \Sigma)$. Using (17), we find that this means that

$$c_2(\nu) = \frac{(\gamma - \mu_2)^2}{2\Sigma_{22}}.$$

Furthermore, for $\nu_i = \mathcal{N}(\boldsymbol{\mu}_i, \Sigma)$ and $\nu_j = \mathcal{N}(\boldsymbol{\mu}_j, \Sigma)$ with $i^*(\{\boldsymbol{\mu}_i, \boldsymbol{\mu}_j\}) = i$, we have

$$c_1(\nu_i, \nu_j, w) = \min_{\substack{\boldsymbol{\lambda}_i, \boldsymbol{\lambda}_j \in \mathbb{R}^2 \\ \lambda_{i,1} \leq \lambda_{j,1} \text{ and } \lambda_{j,2} \leq \gamma}} \frac{1}{2}\|\boldsymbol{\mu}_i - \boldsymbol{\lambda}_i\|_{\Sigma^{-1}}^2 + \frac{w}{2}\|\boldsymbol{\mu}_j - \boldsymbol{\lambda}_j\|_{\Sigma^{-1}}^2$$

The solution for $\boldsymbol{\lambda}_i, \boldsymbol{\lambda}_j$ falls in three cases, depending on which of the two constraints are active at the solution.

1. $\lambda_{i,1} \leq \lambda_{j,1}$ active and $\lambda_{j,2} \leq \gamma$ inactive. Using (17), we need to find

$$\min_{\substack{\lambda_{i,1},\lambda_{j,1} \in \mathbb{R}^2 \\ \lambda_{i,1} \leq \lambda_{j,1}}} \frac{(\mu_{i,1} - \lambda_{i,1})^2}{2\Sigma_{11}} + w\frac{(\mu_{j,1} - \lambda_{j,1})^2}{2\Sigma_{11}},$$

   where we have used

$$\lambda_{i,2} \;=\; \mu_{i,2} - \frac{\Sigma_{12}}{\Sigma_{11}}(\mu_{i,1} - \lambda_{i,1}) \qquad \text{and} \qquad \lambda_{j,2} \;=\; \mu_{j,2} - \frac{\Sigma_{12}}{\Sigma_{11}}(\mu_{j,1} - \lambda_{j,1}).$$

   We then make the two means in the first coordinate equal, and get

$$\lambda_{i,1} = \lambda_{j,1} \;=\; \frac{\frac{1}{w}\Sigma_{11}\mu_{i,1} + \Sigma_{11}\mu_{j,1}}{\frac{1}{w}\Sigma_{11} + \Sigma_{11}}$$

   with value

$$\frac{1}{2}\frac{(\mu_{i,1} - \mu_{j,1})^2}{\Sigma_{11} + \frac{1}{w}\Sigma_{11}}$$

   and optimal second coordinates

$$\lambda_{i,2} \;=\; \mu_{i,2} - \frac{\Sigma_{12}}{\Sigma_{11} + \frac{1}{w}\Sigma_{11}}(\mu_{i,1} - \mu_{j,1}) \qquad \text{and} \qquad \lambda_{j,2} \;=\; \mu_{j,2} + \frac{\frac{1}{w}\Sigma_{12}}{\Sigma_{11} + \frac{1}{w}\Sigma_{11}}(\mu_{i,1} - \mu_{j,1}).$$

2. $\lambda_{i,1} \leq \lambda_{j,1}$ inactive and $\lambda_{j,2} \leq \gamma$ active. Using (17), we find that then $\boldsymbol{\lambda}_i = \boldsymbol{\mu}_i$ and

$$\boldsymbol{\lambda}_j \;=\; \begin{pmatrix} \mu_{j,1} - \frac{\Sigma_{21}}{\Sigma_{22}}(\mu_{j,2} - \gamma) \\ \gamma \end{pmatrix}$$

   and the cost is

$$\frac{w}{2}\frac{(\mu_{j,2} - \gamma)^2}{\Sigma_{22}}.$$

3. both $\lambda_{i,1} \leq \lambda_{j,1}$ and $\lambda_{j,2} \leq \gamma$ active. Here we need to optimize

$$\min_{\theta \in \mathbb{R}^2} \frac{(\mu_{i,1} - \theta)^2}{2\Sigma_{11}} + \frac{w}{2}\frac{\Sigma_{22}(\mu_{j,1} - \theta)^2 - 2\Sigma_{12}(\mu_{j,1} - \theta)(\mu_{j,2} - \gamma) + \Sigma_{11}(\mu_{j,2} - \gamma)^2}{\Sigma_{11}\Sigma_{22} - \Sigma_{12}^2}.$$

   Cancelling the $\theta$ derivative gives

$$\theta \;=\; \frac{(\Sigma_{11}\Sigma_{22} - \Sigma_{12}^2)\mu_{i,1} + w\Sigma_{11}\left(\Sigma_{22}\mu_{j,1} - \Sigma_{12}(\mu_{j,2} - \gamma)\right)}{(\Sigma_{11} + w\Sigma_{11})\Sigma_{22} - \Sigma_{12}^2}.$$

   With that, the value and optimizers become

$$w\frac{\Sigma_{22}(\mu_{i,1} - \mu_{j,1})^2}{2((\Sigma_{11} + w\Sigma_{11})\Sigma_{22} - \Sigma_{12}^2)}$$
$$+ 2w\frac{\Sigma_{12}(\mu_{i,1} - \mu_{j,1})(\mu_{j,2} - \gamma)}{2((\Sigma_{11} + w\Sigma_{11})\Sigma_{22} - \Sigma_{12}^2)}$$
$$+ w\frac{(w\Sigma_{11} + \Sigma_{11})(\mu_{j,2} - \gamma)^2}{2((\Sigma_{11} + w\Sigma_{11})\Sigma_{22} - \Sigma_{12}^2)}$$

   and

$$\boldsymbol{\lambda}_i \;=\; \begin{pmatrix} \theta \\ \mu_{i,2} - \frac{\Sigma_{12}}{\Sigma_{11}}(\mu_{i,1} - \theta) \end{pmatrix} \qquad \text{and} \qquad \boldsymbol{\lambda}_j \;=\; \begin{pmatrix} \theta \\ \gamma \end{pmatrix}.$$

### A.3.2   Proof of proposition 2.3

Let $\nu = \mathcal{N}(\boldsymbol{\mu}, \Sigma), \nu' = \mathcal{N}(\boldsymbol{\lambda}, \Sigma')$ and $\mathcal{M} = \mathcal{M}_G$. To derive the KLinf, we will use the following well-known facts about matrix derivatives:

$$\frac{d}{dA}\log|A| \;=\; (A^{-1})^T \text{ and } \frac{\partial}{\partial A}\operatorname{tr}(A^{-1}B) \;=\; -(A^{-1}BA^{-1})^T.$$

Together with the fact that $\Sigma$ and $\Sigma'$ are symmetric (and so are their inverses), it follows that

$$\frac{\partial}{\partial \Sigma'} \operatorname{KL}\left(\nu \| \nu'\right) = \left(\Sigma'^{-1} - \Sigma'^{-1}\Sigma\Sigma'^{-1} + \Sigma'^{-1}(\boldsymbol{\mu} - \boldsymbol{\lambda})(\boldsymbol{\mu} - \boldsymbol{\lambda})^T\Sigma'^{-1}\right).$$

Setting to zero and multiplying by $\Sigma'$ from both left and right, gives

$$\Sigma' = \Sigma + (\boldsymbol{\mu} - \boldsymbol{\lambda})(\boldsymbol{\mu} - \boldsymbol{\lambda})^T.$$

By the matrix determinant lemma, we have $|\Sigma'| = |\Sigma|\left(1 + (\boldsymbol{\mu} - \boldsymbol{\lambda})^T\Sigma^{-1}(\boldsymbol{\mu} - \boldsymbol{\lambda})\right)$. Substituting everything back in gives

$$\operatorname{KLinf}(\nu, \boldsymbol{\lambda}) = \frac{1}{2}\ln\left(1 + \|\boldsymbol{\mu} - \boldsymbol{\lambda}\|_{\Sigma^{-1}}^2\right)$$

Using (17), it immediately follows that

$$c_2(\nu) = \frac{1}{2}\ln\left(1 + \frac{(\mu_2 - \gamma)^2}{\Sigma_{22}}\right).$$

Next, let $\nu_i = \mathcal{N}(\boldsymbol{\mu}_i, \Sigma_i)$ and $\nu_j = \mathcal{N}(\boldsymbol{\mu}_j, \Sigma_j)$. Then

$$\min_{\substack{\boldsymbol{\lambda}_i, \boldsymbol{\lambda}_j \in \mathbb{R}^2 \\ \lambda_{j,1} > \lambda_{i,1}, \lambda_{j,2} \le \gamma}} \operatorname{KLinf}(\nu_i, \boldsymbol{\lambda}_i) + w\operatorname{KLinf}(\nu_j, \boldsymbol{\lambda}_j)$$

$$= \min_{\theta \in \mathbb{R}} \left(\min_{\substack{\boldsymbol{\lambda}_i \in \mathbb{R}^2 \\ \lambda_{i,1} < \theta}} \operatorname{KLinf}(\nu_i, \boldsymbol{\lambda}_i) + \min_{\substack{\boldsymbol{\lambda}_j \in \mathbb{R}^2 \\ \lambda_{j,1} \ge \theta, \lambda_{j,2} \le \gamma}} w\operatorname{KLinf}(\nu_j, \boldsymbol{\lambda}_j)\right). \tag{18}$$

We have essentially already computed the first term inside the parantheses above; it equals $\frac{1}{2}\ln\left(1 + \frac{(\mu_{i,1} - \theta)_+^2}{\Sigma_{i,11}}\right)$. For the second term, we see

$$\min_{\substack{\boldsymbol{\lambda}_j \in \mathbb{R}^2 \\ \lambda_{j,1} \ge \theta, \lambda_{j,2} \le \gamma}} w\operatorname{KLinf}(\nu_j, \boldsymbol{\lambda}_j) = \begin{cases} 0 & \text{if } \mu_{j,2} \le \gamma \text{ and } \mu_{j,1} \ge \theta \\ \frac{w}{2}\log\left(1 + \frac{(\mu_{j,2} - \gamma)_+^2}{\Sigma_{j,22}}\right) & \text{if } \mu_{j,1} - \frac{\Sigma_{j,12}}{\Sigma_{j,22}}(\mu_{j,2} - \gamma)_+ \ge \theta \\ \frac{w}{2}\log\left(1 + \frac{(\mu_{j,1} - \theta)_-^2}{\Sigma_{j,11}}\right) & \text{if } \mu_{j,2} + \frac{\Sigma_{j,12}}{\Sigma_{j,11}}(\mu_{j,1} - \theta)_- \le \gamma \\ w\operatorname{KLinf}(\nu_j, (\theta, \gamma)) & \text{else.} \end{cases}$$

The values in the second and third case are the result of ignoring one of the two constraints. If the optimizer of this less constrained problem is a feasible solution, the optimizer of the entire problem has been found. If this does not happen for either of them, the optimum must be in the point $\boldsymbol{\lambda}_j = (\theta, \gamma)$ (i.e. both constraints must be active). We'll refer to these cases as case 0-3 respectively.

To get a further handle on this quantity, we will proceed by case distinctions.

1. Let's first consider $\mu_{j,2} \le \gamma$. If $\Sigma_{j,12} < 0$, then $\mu_{j,2} + \frac{\Sigma_{j,12}}{\Sigma_{j,11}}(\mu_{j,1} - \theta)_- \le \gamma$ for all $\theta$, so that the conditions to case 2 will always be satisfied (case 2 coincides with case 0 for $\theta < \mu_{j,1}$). The optimum value of $\theta$ will then be in $[\mu_{j,1}, \mu_{i,1}]$, since the first term in (18) is decreasing and the second term is zero on $(-\infty, \mu_{j,1}]$, and reversed for $[\mu_{i,1}, \infty)$. It follows that the optimum $\theta$ can be found by solving

$$\frac{\mathrm{d}}{\mathrm{d}\theta} \frac{1}{2}\log\left(1 + \frac{(\mu_{i,1} - \theta)^2}{\Sigma_{i,11}}\right) + \frac{w}{2}\log\left(1 + \frac{(\mu_{j,1} - \theta)^2}{\Sigma_{j,11}}\right) = 0$$

$$\frac{-(\mu_{i,1} - \theta)}{\Sigma_{i,11} + (\mu_{i,1} - \theta)^2} + w\frac{\theta - \mu_{j,1}}{\Sigma_{j,11} + (\mu_{j,1} - \theta)^2} = 0, \tag{19}$$

so it is a matter of finding the roots of a cubic (and pruning to $[\mu_{j,1}, \mu_{i,1}]$).

If $\Sigma_{j,12} > 0$, then for $\theta \le \mu_{j,1} + \frac{\Sigma_{j,11}}{\Sigma_{j,12}}(\gamma - \mu_{j,2})$, we will still be in case 2, but we end up in case 3 for $\theta$ larger than that. This does not affect the analysis if $\mu_{i,1} < \mu_{j,1} + \frac{\Sigma_{j,11}}{\Sigma_{j,12}}(\gamma - \mu_{j,2})$. If $\mu_{i,1}$ is

larger than that, we can separately find the minimizer for $\theta \in [\mu_{j,1} + \frac{\Sigma_{j,11}}{\Sigma_{j,12}}(\gamma - \mu_{j,2}), \mu_{i,1}]$. This can be done by solving

$$\frac{\mathrm{d}}{\mathrm{d}\theta} \frac{1}{2} \log\left(1 + \frac{(\mu_{i,1} - \theta)^2}{\Sigma_{i,11}}\right) + w \, \mathrm{KLinf}(\nu_j, (\theta, \gamma)) = 0$$

$$1 \frac{-(\mu_{i,1} - \theta)}{\Sigma_{i,11} + (\mu_{i,1} - \theta)^2} + \frac{w}{|\Sigma_i|} \frac{\Sigma_{j,12}(\mu_{j,2} - \gamma) - \Sigma_{j,22}(\mu_{j,1} - \theta)}{1 + (\boldsymbol{\mu}_j - (\theta, \gamma))^T \Sigma_j^{-1} (\boldsymbol{\mu}_j - (\theta, \gamma))} = 0, \tag{20}$$

which again comes down to solving a cubic (and clipping to the right interval). We can then minimize over the two minima to find the global minimizer.

2. Now we will consider the case $\mu_{j,2} > \gamma$. Then case 0 can never hold and we will be in case 1 for all $\theta \leq \mu_{j,1} - \frac{\Sigma_{j,12}}{\Sigma_{j,22}}(\mu_{j,2} - \gamma)$. If $\Sigma_{j,12} > 0$, case 2 can also never be satisfied, so we will be in case 3 for all $\theta \geq \mu_{j,1} - \frac{\Sigma_{j,12}}{\Sigma_{j,22}}(\mu_{j,2} - \gamma)$. If this bound is larger than $\mu_{i,1}$, then each $\theta \in [\mu_{i,1}, \mu_{j,1} - \frac{\Sigma_{j,12}}{\Sigma_{j,22}}(\mu_{j,2} - \gamma)]$ is a minimizer. If the bound is smaller than $\mu_{i,1}$, we can use (20) to find the optimal value.

If $\Sigma_{j,12} < 0$, then we will only be in case 3 until $\theta \geq \mu_{j,1} - \frac{\Sigma_{j,11}}{\Sigma_{j,12}}(\mu_{j,2} - \gamma)$, at which point we enter case 2. As a sanity check, notice that $-\frac{\Sigma_{j,11}}{\Sigma_{j,12}} > -\frac{\Sigma_{j,12}}{\Sigma_{j,22}}$, since $\Sigma_{j,11}\Sigma_{j,22} > \Sigma_{j,12}^2$ by positive semi-definiteness (so case 2 happens after 3). So we can use (19) to find the minimum over all large $\theta$, and again find the global minimizer by minimizing over the two cases.

### A.4   Proof of Proposition 2.4

First, let $\nu \in \mathcal{M}_B$, then

$$\mathrm{KLinf}(\nu, \boldsymbol{\lambda}) = \min_{\substack{Q \in \triangle_{[0,1]^2} \\ \boldsymbol{m}(Q) = \boldsymbol{\lambda}}} \mathrm{KL}(\nu \| Q). \tag{21}$$

Introducing Lagrange multipliers $d_1, d_2, d_3$, the constraints can be included in the objective as

$$\min_{Q \geq 0} \max_{d_1, d_2, d_3} \mathrm{KL}(\nu \| Q) + d_1(\mathbb{E}_Q[1] - 1) + d_2(\mathbb{E}_Q[X] - \lambda_1) + d_3(\mathbb{E}_Q[Y] - \lambda_2). \tag{22}$$

Here and in the following, $\mathbb{E}_Q[\cdot]$ is meant to be read as $\mathbb{E}_{(X,Y) \sim Q}[\cdot]$. This becomes more tractable by (as is usual for Lagrange multipliers) swapping the max and min, that is,

$$\max_{d_1, d_2, d_3} \min_{Q \geq 0} \mathrm{KL}(\nu \| Q) + d_1(\mathbb{E}_Q[1] - 1) + d_2(\mathbb{E}_Q[X] - \lambda_1) + d_3(\mathbb{E}_Q[Y] - \lambda_2).$$

We show that this swap does not change the value of the problem after further simplifying. The inner minimum has optimizer

$$\frac{dQ}{d\nu}(x, y) = \frac{1}{d_1 + d_2 x + d_3 y}, \tag{23}$$

so the dual problem becomes

$$\max_{\substack{d_1, d_2, d_3: \\ d_1 + d_2 x + d_3 y \geq 0 \\ \text{for } (x,y) \in [0,1]^2}} \mathbb{E}_\nu\left[\ln\left(d_1 + d_2 X + d_3 Y\right)\right] + 1 - d_1 - d_2 \lambda_1 - d_3 \lambda_2. \tag{24}$$

At this point, one can reparameterize by $d_2' = d_2/d_1$ and $d_3' = d_3/d_1$ and set the derivative with respect to $d_1$ to zero. Then, reparameterizing once more to $a_1 = \frac{d_2'}{1 + d_2' \lambda_1 + d_3' \lambda_2}$ and $a_2 = \frac{d_3'}{1 + d_2' \lambda + d_3' \lambda_2}$ gives the desired form

$$\max_{\substack{a_1, a_2: \\ 1 + a_1(x - \lambda_1) + a_2(y - \lambda_2) \geq 0 \\ \text{for } (x,y) \in [0,1]^2}} \mathbb{E}_\nu\left[\ln\left(1 + a_1(x - \lambda_1) + a_2(y - \lambda_2)\right)\right].$$

It remains to show that the min-max swap was allowed. To this end, work backwards from (24) (the max over the dual function) and first use the Lagrange Duality Theorem [Luenberger, 1997, Theorem 1, Section 8.6] to relate the max of the dual function to a minimum of the original function. In doing so, the domain of optimization changes to the dual of the constraint space, that is, the dual of the set of bounded linear functionals on the compact unit square $[0,1]^2$. By Riesz' Representation

Theorem [see e.g. Hartig, 1983], this dual space is equal to the set of finite signed measures on $[0,1]^2$, so that we recover (22).

Next, fix input arm distributions $\nu_i, \nu_j \in \mathcal{M}_B$ and positive weight $w$. The quantity of interest is

$$\min_{\substack{Q_i, Q_j \in \triangle_{[0,1]^2} \\ m_1(Q_i) \leq m_1(Q_j) \\ m_2(Q_j) \leq \gamma}} \mathrm{KL}(\nu_i \| Q_i) + w\, \mathrm{KL}(\nu_j \| Q_j). \tag{25}$$

Introducing Lagrange multipliers $d_1, d_2, d_3, d_4$, we can write this as

$$\max_{d_1, d_2, d_3 \geq 0, d_4 \geq 0} \min_{Q_i, Q_j \geq 0} \mathrm{KL}(\nu_i \| Q_i) + w\, \mathrm{KL}(\nu_j \| Q_j) + d_1 \big( \underset{Q_i}{\mathbb{E}}[1] - 1 \big) + d_2 \big( \underset{Q_j}{\mathbb{E}}[1] - 1 \big)$$
$$+ d_3 \big( \underset{Q_i}{\mathbb{E}}[X] - \underset{Q_j}{\mathbb{E}}[X] \big) + d_4 \big( \underset{Q_j}{\mathbb{E}}[Y] - \gamma \big),$$

where the implicit min-max swap is allowed by the same argument as before. We then find optimizers

$$\frac{dQ_i}{d\nu_i}(x,y) \;=\; \frac{1}{d_1 + d_3 x} \qquad \text{and} \qquad \frac{dQ_j}{d\nu_j}(x,y) \;=\; w \frac{1}{d_2 - d_3 x + d_4 y} \tag{26}$$

and dual problem

$$\max_{\substack{d_1, d_2, d_3 \geq 0, d_4 \geq 0 \\ \ln \text{args} \geq 0}} \underset{\nu_i}{\mathbb{E}} \left[ \ln\left( d_1 + d_3 X \right) \right] + w \underset{\nu_j}{\mathbb{E}} \left[ \ln \frac{d_2 - d_3 X + d_4 Y}{w} \right] + 1 + w - d_1 - d_2 - d_4 \gamma,$$

where the constraint is that both of the arguments in the logarithm are positive on the unit square, i.e., $d_1 + d_3 x \geq 0$ and $d_2 - d_3 x + d_4 y \geq 0$ for all $(x,y) \in [0,1]^2$. The constraint is homogeneous in the (vector of) Lagrange multipliers. At the optimum, unconstrained optimality (i.e. zero derivative) in the $(d_1, d_2, d_3, d_4)$ direction requires

$$d_1 + d_2 + d_4 \gamma \;=\; 1 + w.$$

We can solve this for $d_2$ and end up with

$$\max_{\substack{d_1, d_3 \geq 0, d_4 \geq 0 \\ \ln \text{args} \geq 0}} \underset{\nu_i}{\mathbb{E}} \left[ \ln\left( d_1 + d_3 X \right) \right] + w \underset{\nu_j}{\mathbb{E}} \left[ \ln \frac{1 + w - d_1 - d_3 X + d_4(Y - \gamma)}{w} \right].$$

If we reparameterize by $b_1 = -(d_1 - 1)/w$, $b_2 = -d_3/w$ and $b_3 = d_4/w$, we get

$$\max_{\substack{b_1, b_2 \leq 0, b_3 \geq 0 \\ \ln \text{args} \geq 0}} \underset{\nu_i}{\mathbb{E}} \left[ \ln\left( 1 - w(b_1 + b_2 X) \right) \right] + w \underset{\nu_j}{\mathbb{E}} \left[ \ln\left( 1 + b_1 + b_2 X + b_3(Y - \gamma) \right) \right].$$

For completeness, let us remark that the inner minimizers (23) and (26) are defined as densities w.r.t. the original arm distributions $\nu, \nu_i, \nu_j$. We now discuss how to recover the outer optimal $Q, Q_1, Q_2$ for the primal problem (21) or (25). The above densities (when plugging in the optimal values of the Lagrange multipliers $\boldsymbol{a}$ or $\boldsymbol{d}$) are part of the answer. However, these densities themselves may not yet sum to one. The reason is that the primal solutions sometimes put mass outside of the support of their corresponding arm distribution. In some cases this is the only way to satisfy the constraints, in other cases it may be driven by optimality. Solving for satisfaction of the primal constraints (i.e. normalization and means) then resolves how the missing mass must be allocated to recover the primal feasible solutions. It is always possible to do so adding mass in a single point on the boundary of the unit square.

# B Proofs for Section 3

## B.1 Proof of Theorem 3.1 (Thresholds)

Let $\boldsymbol{\nu}$ be a bandit with answer $i^* = i^*(\boldsymbol{\nu})$ and true means $\boldsymbol{m}_i = m(\nu_i)$. Following the proof of [Kaufmann and Koolen, 2021, Proposition 21], we have

$$\mathbb{P}_{\boldsymbol{\nu}} \{\tau < \infty \text{ and } \hat{\imath} \neq i^*\}$$
$$\leq \mathbb{P}_{\boldsymbol{\nu}} \{\exists n : i^*(\hat{\boldsymbol{\nu}}(n)) \neq i^* \text{ and } \Lambda_n \geq \beta(\delta, n)\}$$
$$\leq \begin{cases} \mathbb{P}_{\boldsymbol{\nu}} \{\exists n, j \neq i^* : N_{i^*}(n) \,\mathrm{KLinf}(\hat{\nu}_{i^*}(n), \boldsymbol{m}_{i^*}) + N_j(n) \,\mathrm{KLinf}(\hat{\nu}_j(n), \boldsymbol{m}_j) \;\geq\; \beta(\delta, n)\} \\ \mathbb{P}_{\boldsymbol{\nu}} \{\exists n, j : N_j(n) \,\mathrm{KLinf}(\hat{\nu}_j(n), \boldsymbol{m}_j) \;\geq\; \beta(\delta, n)\} \end{cases}$$

where the first case is for $i^* \neq \texttt{None}$ and the second is for $i^* = \texttt{None}$. Furthermore, the last step uses that $\neg i^*$ is covered by motions of at most two arms, one being $i^*$ whenever $i^* \neq \texttt{None}$, by Proposition 2.1. Note that for $i \neq \texttt{None}$ the set $\neg i^*$ has a motion of just arm $i^*$ as well, but the cost of that is subsumed by that of any two-arm deviation. To find $\beta$ that ensures $\delta$-correctness, we need an anytime deviation inequality for a sum of two statistics of the form $N_i(n) \operatorname{KLinf}(\hat{\nu}_i(n), \boldsymbol{m}(\nu_i))$.

For the fixed covarance Gaussian case, we can leverage [Kaufmann and Koolen, 2021, Theorem 9]. Even though that theorem is formulated for arbitrarily many 1d Gaussian arms, we note that for i.i.d. $X_i \sim \mathcal{N}(\boldsymbol{\mu}, \Sigma)$, our $n \operatorname{KLinf}, \frac{n}{2}\|\hat{\boldsymbol{\mu}}(n) - \boldsymbol{\mu}\|^2_{\Sigma^{-1}}$, is a sum of two independent 1d Gaussian contributions. To be more precise, let $Y_i = \Sigma^{-1/2}X_i$, where $\Sigma^{-1/2}$ is the inverse of the positive definite and symmetric square root of $\Sigma$. Then $Y_i \sim \mathcal{N}(\Sigma^{-1/2}\boldsymbol{\mu}, I(2))$, so that $Y_{i,j} \sim \mathcal{N}((\Sigma^{-1/2}\boldsymbol{\mu})_j, 1)$ for $j \in \{1, 2\}$ independently from one another. Furthermore, define $\hat{Y}_n = \frac{1}{n}\sum_{i=1}^n Y_i$. We then see that

$$\operatorname{KL}(\hat{Y}_{n,1}, (\Sigma^{-1/2}\boldsymbol{\mu})_1) + \operatorname{KL}(\hat{Y}_{n,2}, (\Sigma^{-1/2}\boldsymbol{\mu})_2) \; = \; \frac{1}{2}\|\hat{Y}_n - \Sigma^{-1/2}\boldsymbol{\mu}\|^2 \; = \; \frac{1}{2}\|\hat{\boldsymbol{\mu}}(n) - \boldsymbol{\mu}\|^2_{\Sigma^{-1}}.$$

The concentration result by Kaufmann and Koolen [2021, Theorem 9] is stated for sums of univariate KLs, as in the left-hand side; this equality allows us to also use it in the fixed covariance setting. We thus find that taking $\beta(\delta, n) = \ln \frac{K}{\delta} + 4\ln \frac{\ln \frac{K}{\delta}}{4} + 8\ln(4 + \ln n/2)$ suffices once $\ln \frac{1}{\delta} \geq 6$. We chose that latter threshold for readability, see Kaufmann and Koolen [2021] for a more involved threshold that works for any $\delta \in (0, 1)$.

For the unknown covariance, we exploit the expression for KLinf from (11). Recall that for the unknown covariance case we use as our estimate $\hat{\nu}(n) = \mathcal{N}(\hat{\boldsymbol{\mu}}(n), \hat{\Sigma}(n))$, that is, a Gaussian with the maximum likelihood mean and covariance. Then for i.i.d. $X_i \sim \nu = \mathcal{N}(\boldsymbol{\mu}, \Sigma)$,

$$n \operatorname{KLinf}(\hat{\nu}(n), \boldsymbol{\mu}) \; = \; \frac{n}{2}\ln\left(1 + \|\hat{\boldsymbol{\mu}}(n) - \boldsymbol{\mu}\|^2_{\hat{\Sigma}(n)^{-1}}\right).$$

Under $\nu$, the statistic $(n-1)\|\hat{\boldsymbol{\mu}}(n) - \boldsymbol{\mu}\|^2_{\hat{\Sigma}(n)^{-1}}$ has a Hotelling $t^2$ distribution with $n-1$ degrees of freedom in 2 dimensions, and therefore $(n-2)\operatorname{KLinf}(\hat{\nu}(n), \boldsymbol{\mu}) \sim \frac{1}{2}\chi^2_2$. To show this, we will (1) rely on known results on the relations between different distributions and (2) slightly abuse notation to denote distributions in equations instead of random variable. Up to scaling, the statistic we are concerned with is $\ln\left(1 + \frac{1}{n-1}T^2(2, n-2)\right)$. It can be shown that this is the same as $\ln\left(1 + \frac{2}{n-2}F(2, n-2)\right)$ (F-distribution). This, in turn, is equivalent to $\ln\left(1 + \beta'\left(1, \frac{n-2}{2}\right)\right)$ (beta-prime). This can be written as $-\ln\left(\left(1 + \beta'\left(1, \frac{n-2}{2}\right)\right)^{-1}\right) = -\ln\left(\beta'\left(\frac{n-2}{2}, 1\right)\left(\beta'\left(\frac{n-2}{2}, 1\right) + 1\right)^{-1}\right)$, which equals $-\ln\left(\beta\left(\frac{n-2}{2}, 1\right)\right)$ (beta), which is known to be $\exp\left(\frac{n-2}{2}\right)$ (exponential). The desired result follows by reintroducing the scaling factor $\frac{n-2}{2}$.

Hence, for fixed sample size $n$ and $\lambda \in [0, 1]$ the MGF evaluates to $\mathbb{E}_\nu[e^{\lambda(n-2)\operatorname{KLinf}(\hat{\nu}(n), \boldsymbol{\mu})}] = \frac{1}{1-\lambda}$ (alternatively, the above reasoning can be sidestepped by computing this integral directly). We then find that for fixed sample sizes $n_i$ and $n_j$, and threshold $C \geq 2$, we get, by a Chernoff bound,

$$\mathbb{P}_{\boldsymbol{\mu}}\left\{(n_i - 2)\operatorname{KLinf}(\hat{\nu}_i(n_i), \boldsymbol{\mu}_i) + (n_j - 2)\operatorname{KLinf}(\hat{\nu}_j(n_j), \boldsymbol{\mu}_j) \geq C\right\} \; \leq \; \frac{1}{4}C^2 e^{2-C}.$$

We have, with $W_{-1}$ denoting the negative branch of the Lambert function,

$$\frac{1}{4}C^2 e^{2-C} \; = \; \delta \qquad \text{iff} \qquad C \; = \; -2W_{-1}\left(-e^{-1+\frac{1}{2}\ln\delta}\right) \approx \ln\frac{1}{\delta} + 2\ln\ln\frac{1}{\delta}.$$

A weighted union bound over all possible values of $n_i$ and $n_j$ with each prior $\pi$ over $\mathbb{N}$ gives

$$\mathbb{P}_{\boldsymbol{\mu}}\left\{\exists n: \sum_{k \in \{i, j\}}(N_k(n) - 2)\operatorname{KLinf}(\hat{\nu}_k(n), \boldsymbol{\mu}_k) \geq -2W_{-1}\left(-e^{-1+\frac{1}{2}\ln(\delta\pi(N_i(n))\pi(N_j(n)))}\right)\right\} \; \leq \; \delta.$$

Picking $\pi(n) \propto \frac{1}{n(\ln n)^2}$ motivates the choice $\beta(\delta, n) = \ln\frac{K}{\delta} + 2\ln n + 4\ln\ln n + 2\ln(\ln\frac{K}{\delta} + 2\ln t + 4\ln\ln n)$. Notice that there both factors $N_k(n)$ are now replaced by $N_k(n) - 2$. Technically, this could be compensated for by adjusting the threshold. However, this compensation would disappear as the sample sizes grow large, so that it does not matter for any asymptotic arguments. Furthermore,

this is the same as simply using the statistic with factor $N_k(n) - 2$ and noting that, in the limit, it is indistuingishable from our original statistic. In our experiments, we choose to do the latter.

For the bounded case we can use our dual expression from (12) as a maximum over parameters in the compact set $\mathcal{R}_{\boldsymbol{\lambda}} \subseteq \mathbb{R}^2$. Using the technique of [Agrawal et al., 2021, Lemma F.1] based on worst-case regret bounds for online learning with exp-concave losses, we find that for each arm $i$ there is a $\nu_i$ martingale $(M_{i,n})_{n \geq 0}$ (of mixture form) satisfying $M_{i,n} \geq e^{N_i(n) \, \mathrm{KLinf}(\hat{\nu}_i(n), \boldsymbol{m}(\nu_i)) - 1 - 2 \ln(1 + N_i(n))}$. Taking the product over two arms, applying Ville's inequality, and using concavity of the $\ln$ and $N_i(n) + N_j(n) \leq n$ yields $\delta$-correctness for the choice $\beta(\delta, n) = \ln \frac{K}{\delta} + 2 + 4 \ln(1 + n/2)$.

### B.2 Proof of Theorem 3.2 (Asymptotic Optimality)

We show that our algorithm, which consists of (a) the sampling rule (Track-and-Stop with C-tracking) combined with (b) the GLR stopping rule and (c) the empirical-best recommendation rule ensures asymptotic optimality, in the sense that for every bandit $\boldsymbol{\nu}$ with a unique best answer $i^* = i^*(\boldsymbol{\nu})$, our algorithm is $\delta$-correct on $\boldsymbol{\nu}$ and ensures

$$\lim_{\delta \to 0} \frac{\mathbb{E}_{\boldsymbol{\nu}}[\tau_\delta]}{\ln \frac{1}{\delta}} \; = \; T^*(\boldsymbol{\nu}).$$

This argument was pioneered by Garivier and Kaufmann [2016] for BAI in exponential families, and extended to general answers by [Kaufmann and Koolen, 2021, Theorem 17]. [Degenne and Koolen, 2019, Theorem 7] proved the (upper-hemi) continuity assumption which was assumed before, and [Agrawal et al., 2020, Section 6] generalized to non-parametric arms.

To apply the argument for constrained BAI in the three models, we need to check two things. (1) the estimates $\hat{\boldsymbol{\nu}}(t)$ concentrate sufficiently fast around the true bandit $\boldsymbol{\nu}$. (2) the oracle weights $\boldsymbol{\nu} \mapsto \boldsymbol{w}^*(\boldsymbol{\nu})$ are a continuous function of the bandit.

Sufficiently fast concentration of empirical mean and variance in sup norm is argued by [Jourdan et al., 2023, Section H.1.1] for one dimension, and it generalizes to our Gaussian cases in two dimensions. For the bounded model, sufficiently fast concentration of the empirical distribution in Lévy metric is given by the multivariate DKW inequality Naaman [2021].

Continuity of the oracle weights $\boldsymbol{w}^*(\boldsymbol{\nu})$ as a function of the bandit $\boldsymbol{\nu}$ follows from two nested applications of Berge's Theorem, which bottom out in continuity of KLinf. For the Gaussian cases (10) and (11) this holds by inspection. The bounded case (12) requires a small argument.

**Proposition B.1.** *Let $M = [0,1]^2$ be the unit square, and let $M^0 = (0,1)^2$ be its interior. Let $\mathcal{L}$ be the set of all probability distributions on $M$. The function $\mathrm{KLinf}$ from (2) is jointly continuous on $\mathcal{L} \times M^0$, where we equip $\mathcal{L}$ with the Lévy metric.*

*Proof.* For joint lower semi-continuity we can directly apply [Agrawal et al., 2020, Lemma C.2] after observing that our $\mathcal{L}$ is compact. For joint upper semi-continuity we exploit the dual representation (12) and go through [Agrawal et al., 2020, Lemma C.3], noting that Skorokhod's theorem applies to the metric space $M$. □

With those details supplied, the remainder of the classic proof applies. See Kaufmann and Koolen [2021, Appendix D].

Let us sketch the template here for completeness. First,

- Forced exploration ensures that the estimates converge to the true bandit, i.e. $\hat{\boldsymbol{\nu}}(t) \to \boldsymbol{\nu}$. Note that convergence is measured in the appropriate metric, which is Euclidean distance between parameters for the Gaussian cases and Lèvy for the bounded non-parametric case.

- Continuity of the oracle weight map $\boldsymbol{\nu} \mapsto \boldsymbol{w}^*(\boldsymbol{\nu})$ ensures that the sampling weights converge to the oracle weights, i.e. $\boldsymbol{w}_t = \boldsymbol{w}^*(\hat{\boldsymbol{\nu}}(t)) \to \boldsymbol{w}^*(\boldsymbol{\nu})$. Here continuity is in the upper-hemicontinuous sense.

- C-Tracking ensures that the sample counts converge to the sampling weights, i.e. $\frac{\boldsymbol{N}(t)}{t} \to \frac{\sum_{s=1}^t \boldsymbol{w}_s}{t}$. Since the latter average converges to the oracle weights, so do the sampling proportions themselves $\frac{\boldsymbol{N}(t)}{t} \to \boldsymbol{w}^*(\boldsymbol{\nu})$.

- The stopping rule involves the empirical sampling proportion $\frac{N(t)}{t}$ and the estimates $\hat{\nu}(t)$. If these are close to $w^*(\nu)$ and $\nu$ respectively, then the stopping rule kicks in around time $T^*(\nu) \ln \frac{1}{\delta}$.

Without quantifying all these convergences, we are proving $\mathbb{P}_\nu \left\{ \lim_{\delta \to 0} \frac{\tau_\delta}{\ln \frac{1}{\delta}} = T^*(\nu) \right\} = 1$. The crux of the in-expectation argument is to invoke *sufficiently fast concentration* of the estimates, to ensure that the contribution to the expected sample complexity due to failures of any of the above convergences is of lower-order in $\ln \frac{1}{\delta}$.

