# OpenReview forum: "Constrained Best Arm Identification"
_NeurIPS.cc/2025/Conference — NeurIPS 2025 poster_

### Official Review · Reviewer_gFFA · 2025-06-14

**Clarity:** 1
**Significance:** 2
**Originality:** 2
**Rating:** 2
**Confidence:** 3

**Summary:**

The authors study a fixed confidence finite-arm pure exploration problem where each arm is related to a bivariate distribution over rewards and costs. The goal of the learner is to identify the arm with the largest expected reward subject to satisfy an hard constraint on its cost.

They assume that such an optimal arm is unique and they derive, as usual in the literature, a lower bound on the sample complexity for achieving the aforementioned objective. Then, they propose a track-and-stop (GK16) procedure to solve the problem. Specifically, this procedure can be implemented in the three following scenarios:
- Gaussian distributions with fixed covariance
- Gaussian distributions with unknown covariance
- Non-parametric distributions on the unit square.

The authors then run some numerical experiments and conclude the paper by studing the impact of the dependence on the sample complexity.

**Questions:**

As a suggestion, I invite the authors to improve the clarity of the manuscripts (see problems mentioned above). To gain eventual space, it might also be useful to condense the known and unknown covariance setting.

**Ethical Concerns:**

["NO or VERY MINOR ethics concerns only"]

**Final Justification:**

I am confirming my score for the reasons highlighted during the discussion phase

**Limitations:**

See weaknesses above.

This paper is mainly of a theoretical nature. I do not see a direct path to negative societal concerns.

**Paper Formatting Concerns:**

None.

**Quality:**

2

**Strengths And Weaknesses:**

# Strengths

The problem studied by the authors looks interesting as in some applications there is indeed this notion of costs to be taken into account.
To improve the state-of-the-art, I also suggest the authors to include the following references:
- "Optimal Multi-Fidelity Best-Arm Identification", Poiani et al., NeurIPS 2024
- "Cost Aware Best Arm Identification", Kanarios et al., 2024
as they are related to this problem of dealing with costs in real-world scenarios.

# Weaknesses
- The paper has some clarity issues that make the read hard in some points. For instance: **(1)**  Lines 66-68 are not clear to me. **(2)** Furthermore, it is probably missing something in lines 119-123. This is complicating the read (also lambda^* not defined in lines 120-122). **(3)** In Figure 3 I fail to see T^* anywhere. **(4)**

- Large part of this paper deals with the computation of T^* which the authors claims to be mandatory (lines 93-97) but which is not that necessary (see also my comment below). The arguments behind this points adopts the reasoning of GK16 with the additional linear constraint on the costs. This part of the paper seems to be technically a mild contribution to the pure exploration literature.

- The algorithmic part does not present innovation in the sampling rule. Here, novel arguments only arises in proving the correctness of the method given that the feedback is different wrt standard works. However, the proof of Theorem 1 is really poorly detailed and informal and I cannot ensure its correctness,

- Other (minor) concern. The discussion in lines 93-97 is inaccurate as there are several works that do not explicitly compute T^* nor w^* in practice. Examples include "Non-asymptotic pure exploration by solving games", Degenne et al., NeurIPS 2019 and "Fast Pure Exploration via Frank Wolfe", Wang et al., NeurIPS 2021.  It is furthermore not clear why the authors are referring to the regret minimization method of Degenne et al., 2019 as GA-TaS, since the two algorithms are not really working under the same principles.

---

> ### Author Rebuttal · Authors · 2025-07-31
>
> We thank the reviewer for the careful reading and the concrete pointers to literature. Below, we address the questions and concerns raised. For clarity, reviewer comments are repeated in **_bold italics_**, followed by our responses in regular font.
>
> ## Strengths ##
>
> **_The problem studied by the authors looks interesting as in some applications there is indeed this notion of costs to be taken into account. To improve the state-of-the-art, I also suggest the authors to include the following references:[...]_**
>
> These are indeed interesting references that are worth including. To clarify how they differ from our setting: in the first work, the cost is known in advance and tied to the fidelity level, rather than being an intrinsic property of the arm itself. Similarly, in the second, the cost is independent of the reward, and the objective remains to identify the (unconstrained) arm with the highest reward, while minimizing the overall sampling cost. In contrast, our setting imposes a constraint on the arm selection itself, rather than on the sampling process. Finally, both papers focus exclusively on exponential families, whereas our framework does not make this restriction.
>
> ## Weaknesses ##
>
> **_The paper has some clarity issues that make the read hard in some points. For instance: (1) Lines 66-68 are not clear to me. (2) Furthermore, it is probably missing something in lines 119-123. This is complicating the read (also $\\lambda^\\star$ not defined in lines 120-122). (3) In Figure 3 I fail to see $T^\\star$ anywhere.(4)_**
>
> We agree that certain parts of the paper could be clearer, and we appreciate the specific pointers. We will revise these parts to improve readability. As for point (3): the caption and legend are currently lacking in clarity. All of the lines in the left plot are expected stopping times computed with different weights, as indicated by the color of the line. The blue line is the expected stopping time with optimal weights, which corresponds to the characteristic time.
>
> **_Large part of this paper deals with the computation of $T^\\star$ which the authors claims to be mandatory (lines 93-97) but which is not that necessary (see also my comment below). The arguments behind this points adopts the reasoning of GK16 with the additional linear constraint on the costs. This part of the paper seems to be technically a mild contribution to the pure exploration literature._**
>
> As for the first part: we will rephrase those sentences, we had not meant to phrase it as the only way to go about the problem. However, one of the reasons we are interested in computing $T^\\star$ is to characterize the difficulty of the problem, which allows us to e.g. investigate the effect of dependence on said difficulty. We agree that it is, strictly speaking, not needed for the algorithm itself, even though that is the approach we take here.
>
> Furthermore, we disagree that it is a mild contribution to the pure exploration literature. We solve a practical and intricate problem. In particular, considering the feasibility constraint introduces two transportation cost functions, for which finding the infimum requires sub-casing, a new answer “none”, a different GLRT, and new thresholds.
>
> **_The algorithmic part does not present innovation in the sampling rule. Here, novel arguments only arises in proving the correctness of the method given that the feedback is different wrt standard works. However, the proof of Theorem 1 is really poorly detailed and informal and I cannot ensure its correctness_**
>
> As we also noted in our response to reviewer MhZm, we understand the concern and acknowledge that we were not explicit enough  in justifying the applicability of results from prior work to our setting. We will revise the proofs to and elaborate on the intermediate steps where appropriate, to make the argument more self-contained and accessible.
>
> **_Other (minor) concern. The discussion in lines 93-97 is inaccurate as there are several works that do not explicitly compute $T^\\star$ nor $w^\\star$ in practice. Examples include "Non-asymptotic pure exploration by solving games", Degenne et al., NeurIPS 2019 and "Fast Pure Exploration via Frank Wolfe", Wang et al., NeurIPS 2021. It is furthermore not clear why the authors are referring to the regret minimization method of Degenne et al., 2019 as GA-TaS, since the two algorithms are not really working under the same principles._**
>
> For the first part, we refer to our response to the point above. Regarding the citation, we agree that its placement was suboptimal. Our intention was to acknowledge that our gradient ascent approach, used to address the high computational cost in the bounded case, was inspired by a method used by Degenne et al. (2019). We will revise the text to make this connection clearer.
>
> ## Questions ##
>
> **_As a suggestion, I invite the authors to improve the clarity of the manuscripts (see problems mentioned above). To gain eventual space, it might also be useful to condense the known and unknown covariance setting._**
>
> We agree with the concerns about clarity and will address them through the following improvements:
> 1. more clearly highlighting the key challenges of the problem;
> 2. making our contributions more prominent and easier to identify; and
> 3. expanding on the less obvious steps in the proofs to improve readability and self-containment.

---

> > ### Comment · Reviewer_gFFA · 2025-08-05
> > **Ack.**
> >
> > I thank the authors for their rebuttal and the other reviewers for their feedback.
> >
> > Given the rebuttal, I retain that this work needs to go through an important revision to fix several clarity issues, together with formally proving the correctness of some of the results. I maintain my score as I believe that another round of the review process could benefit this paper significantly.

---

> > > ### Author Response · Authors · 2025-08-06
> > >
> > > We thank the reviewer for their time and feedback. We will carefully address the concerns raised in the initial review during the revision process.

---

### Official Review · Reviewer_MhZm · 2025-06-30

**Clarity:** 2
**Significance:** 3
**Originality:** 2
**Rating:** 5
**Confidence:** 4

**Summary:**

This paper studies best-arm identification (BAI) under an additional constraint in the fixed-confidence setting. Specifically, the learner observes bivariate outcomes and aims to identify, with high probability, the arm that maximizes the first criterion, subject to the second criterion being below a specified threshold. The authors consider several distributional assumptions over the arms: bivariate normal distributions with known covariance, bivariate normal distributions with unknown covariance, and general bivariate distributions supported on $[0,1]^2$. An information-theoretic lower bound on the sample complexity is derived, and an adaptation of the Track-and-Stop algorithm (Garivier and Kaufmann 2016) is proposed to match this lower bound in the small error regime.

**Questions:**

**Minor comments**:
* l.91 $KL(\delta\\| 1- \delta)$ is conflicting with earlier notation
* The standard errors reported in Section 4 are surprisingly small given the  confidence level and the sample complexities reported. It would help if the authors reported the (co)variances used in these experiments
**Questions** :
* Why does the "Oracle" strategy perform significantly worse than the other algorithms in some experiments?
* As the authors assume common covariance in the Gaussian setting. How is the MLE of covariance computed ?
* How were the oracle allocation weights computed?
* In Figure 3, negative correlation does not seem to influence the sample complexity. Can the authors provide some intuition for this?
* Does Figure 3 suggest a regime of lower sample complexity for uniform sampling (which by nature ignores correlation) than the optimal algorithm that ignores correlation ?

**Ethical Concerns:**

["NO or VERY MINOR ethics concerns only"]

**Final Justification:**

My initial review raised some technical concerns that I felt required more rigorous justification. The authors have since provided convincing arguments addressing these points. The remaining issues noted in my rebuttal appear to be addressable in a revised version without requiring substantial changes to the structure of the manuscript. Overall, the paper presents several interesting contributions ; notably, the concentration inequalities (Theorem 3.1) and the analysis of how correlation affects sample complexity. While the algorithm itself is another variant of the Track-and-Stop framework, I commend the authors for their efforts in designing computationally efficient methods for characterizing and computing the oracle weights (see Theorem 2.2 and Propositions 2.2–2.4).

**Limitations:**

yes, this work is mainly theoretical

**Paper Formatting Concerns:**

No formatting concerns

**Quality:**

3

**Strengths And Weaknesses:**

**Strengths**:
The paper tackles a well-motivated and practically relevant problem with compelling applications.
An information-theoretic lower bound is stated and the authors proposed an algorithm to match this bound in the small error regime. We  appreciate the experimental results provided to support the theoretical claims and the ablation studies on the effect of correlation. The computational cost of the proposed algorithm is discussed and enhances the practical perspective of the work. However the papers contains major weaknesses and some soundness issues as stated below.

**Weaknesses**:
* The related work section overlooks several contributions on constrained pure exploration, including works that studied the same problem as the authors. For example  Katz-Samuels and Scott, 2019, Top Feasible Arm Identification) studied the identification of the best arm under linear feasibility constraints (which generalizes  the constrained BAI  problem to dimension $d$); Faizal and Nair, 2022 (Constrained Pure Exploration Multi-Armed Bandits with a Fixed Budget) studied the same problem in the fixed-budget setting;, Kone et al 2025 (Constrained Pareto Set Identification with Bandit Feedback) or Katz-Samuels and Scott (Feasible Arm Identification)
* The novelty is limited as it is another direct application of track-and-stop methodology. The approach relies heavily on a relatively straightforward adaptation of the Track-and-Stop methodology. As such, the novelty appears incremental. The computational cost is also prohibitive compared to simpler approaches like in  Katz-Samuels and Scott, 2019,  or even compared to Racing approaches (as reported in Section 4)
* The proof of theorem 3.1 is not correct :  for the setting with known covariance, statement from l.500 to l.502 is not correct. The authors incorrectly references Kaufmann and Koolen (2021), which addresses exponential families with real-valued parameters. The result does not directly apply to the bivariate case with dependent marginals. Appropriate references for this setting include section 4.1.3
of Degenne 2019 (Impact de la structure sur la conception et l’analyse d’algorithmes de bandits)
 or Lemma 20 in Kone et al 2025 (Pareto Set Identification With Posterior Sampling). For the setting with unknown  covariance, statement on line l.510 is not correct, $\frac{n-2}{2} \log(1+ \\| \hat \mu_n - \mu \\|_{\hat \Sigma_n^{-1}}^{2})$ is not a  $\chi_2^2/2$, hence the derived concentration result is not correct.
* The paper refers to prior work (Garivier and Kaufmann, 2016; Jourdan et al., 2023) for the analysis of Track-and-Stop. However, those works deal with univariate settings and do not address the concentration properties of empirical mean vectors and covariances required here. Since this is a central contribution of the paper, the analysis should be made self-contained.
* The authors propose using D-tracking for arm allocation, but as pointed out by Degenne and Koolen 2019, D-tracking may fail to converge when the optimal allocation is not unique. The paper does not justify the uniqueness of the optimal allocation in the settings studied.

---

> ### Author Rebuttal · Authors · 2025-07-31
>
> We thank the reviewer for acknowledging the theoretical and practical impact of our work. We appreciate the careful reading, the concrete pointers to missed references and the insightful questions. Below, we address the questions and concerns raised in order (we do not repeat the reviewers comments due to a lack of space).
>
> ## Weaknesses ##
>
> - All of the mentioned articles are indeed closely related to our setting, and we will be sure to include appropriate references. The most relevant among them is perhaps Katz-Samuels and Scott, 2019, Top Feasible Arm Identification. We would like to highlight a few key differences: (1) their work focuses exclusively on the sub-Gaussian setting, while we introduce a framework capable of handling arbitrary models. This includes, as we demonstrate, non-parametric settings such as distributions with a common rectangular support; and (2) our framework allows us to explicitly investigate the effect of dependence between the cost and reward on algorithm performance.
>
> - 1. While we can use the transportation lemma, finding the optimal sampling proportions is not straightforward. So the fact that it is formalized in the TaS framework, does not mean it is ‘direct’. The TaS framework allows us to use the notion of characteristic time and the transportation lemma, but every new pure exploration problem (pep), or even a different model for the same pep (gaussian with unknown variance, bounded support, etc) would yield different transportation cost function and require a potentially different solution (if it even exists) for the oracle weights. In the constraint BAI, the feasibility constraint(s) introduces two transportation cost functions, for which finding the infimum requires a sub-casing, a new answer “none”, new GLRT thresholds, and an efficient algorithm to estimate the oracle weights.
>     2. We wanted to provide a common computational interface for all models considered in this paper, mainly motivated by the bounded setting, which can be well described in information theoretic terms. In this regard, we were especially inspired by previous works such as by Agrawal, Juneja, and Glynn (Optimal δ-correct best-arm selection for heavy-tailed distributions) and Agrawal, Koolen, and Juneja (Optimal Best Arm Identification Methods for Tail-Risk Measures), which illustrate the effectiveness of information-theoretic tools in non-parametric settings.
>     3. The algorithm is not computationally prohibitive, as we show in Theorem 2.1 it can be solved by a nested binary search. Especially the Gaussian case (unknown variance) requires just solving a cubic equation, which is very fast. The non parametric case is slower than the gaussian models, but the bottleneck is dealing with the empirical data distribution, which is enhanced by the nested search (a few milliseconds per instance). To avoid the nested search, we propose a gradient-ascent technique, which is extremely fast. (It is hard to benchmark our computational performance, because related work on constrained BAI does not handle non parametric cases)
>
> - We respond to the two points that are raised separately:
>
>    * l. 500 to l.502: While we agree that the setting studied by Kaufmann and Koolen (2021) does not directly apply to our setting, we do not agree that this means that their result is not applicable at all. The way in which we implicitly use it is as follows: define $Y_n = \\Sigma^{-1/2}X_n$ where $\\Sigma^{-1/2}$ is the inverse of the positive definite and symmetric square root of $\Sigma$. We then have $Y_n \\sim N(\\Sigma^{-1/2}\\mu, I_2)$, so that $Y_{n,j}\\sim N((\\Sigma^{-1/2}\mu)_j, 1)$ for $j=1,2$  independently from one another. Let us also denote $\\hat Y_n = \\frac{1}{n} \\sum\_{i=1}^n Y_i$. Theorem 9 by Kaufmann and Koolen (2021) can then be used to gain control over sums of terms of the form $nd(\\hat Y\_{n,1}, (\\Sigma^{-1/2}\\mu)\_1) +nd(\\hat Y\_{n,2}, (\\Sigma^{-1/2}\\mu)\_2)$. That is, each arm gives us two univariate Gaussian contributions. These terms can be rewritten to $n/2 \\| \hat Y_n - \\Sigma^{-1/2}\\mu\\|^2 =n/2 \\| \\hat Y_n - \\Sigma^{-1/2}\\mu\\|^2= n/2 \\|\\hat \\mu - \\mu \\|\_{\\Sigma^{-1}}$, i.e., precisely the $n$KLinf that we consider here. However, we now see that this reasoning was very implicit and that it should have been made more explicit.
>
>    * Unknown covariance: we politely disagree, but admit that the argument could have been made more concrete. To show the validity, we will (1) rely on known results on the relations between different distributions and (2) slightly abuse notation to denote distributions in equations instead of random variables. The statistic we consider is $\\ln (1+\\frac{T^2(2,n-2)}{n-1})$ (Hotelling's T-squared) scaled by $\\frac{n-2}{2}$, but we drop the scaling for now. The former is equivalent to $\\ln (1+\\frac{2}{n-2} F(2,n-2))$ (F-distribution), which in turn is the same as $\\ln (1+\\beta'(1, \\frac{n-2}{2}))$ (Beta prime). Now, write this as $-\\ln (1/(1+\\beta'(1,\\frac{n-2}{2})))= -\\ln ( \\beta'(\\frac{n-2}{2},1)/(\\beta'(\\frac{n-2}{2},1)+1))$ which equals $-\\ln (\\beta(\\frac{n-2}{2},1))$ (Beta). This is known to be equivalent to $\\exp(\\frac{n-2}{2})$. After reintroducing the scaling factor of $\\frac{n-2}{2}$, we get $\\frac{1}{2} \\exp(1/2)=\\frac{1}{2} \\chi^2(2)$, as was to be shown.
>
> - We understand the concern and acknowledge that we were somewhat non-verbose in invoking results from prior work without fully explaining why they apply in our setting. We will revise the text to clarify these connections and elaborate on the intermediate steps where needed to make the argument more self-contained and accessible.
>
> - This is a valid concern. While it is true that D-tracking may face convergence issues in cases where the optimal weights are non-unique, we note that this does not affect the $\\delta$-correctness of the algorithm. Moreover, in all our simulations, D-tracking consistently converged and performed as expected, suggesting that this issue does not arise in practice for the settings we consider.
>
> ## Minor Comments ##
>
> - To the best of our knowledge, this is the first mention of KL in the paper, so we are unsure about the conflict being referenced. That said, we agree that the notation is never formally introduced, and we will make sure to define it clearly.
>
> - All the bandits used in the simulations are shown in Figure 2, along with the corresponding covariance matrix. The error bars represent standard errors of the average runtimes, which we will clarify in the text. As expected, the standard errors decrease as the number of simulations increases.
>
> ## Questions ##
>
> - This is an intriguing question that we unfortunately do not have a more satisfactory answer to than “it happens in all pure exploration experiments”, see e.g. Degenne, Koolen, and Ménard (Non-Asymptotic Pure Exploration by Solving Games, 2019) and Jourdan, Degenne, and Kaufmann (Dealing with Unknown Variances in Best-Arm Identification, 2023). However, it is definitely worth mentioning in the discussion of the results.
>
> - In the Gaussian setting, we consider two models: (1) the covariance matrix is fixed and shared across all arms, and (2) each arm has its own unknown covariance matrix, which may differ across arms. In the first case, we use the true covariance directly and do not estimate it. In the second case, the covariance is estimated via the standard MLE (that is, a scaled version of the sample covariance) for each arm separately without imposing any additional assumptions. To avoid singularity issues, we always start with at least three data points.
>
> - The oracle allocation weights are computed under the assumption of oracle knowledge of which bandit we are in. Specifically, the optimal weights from Theorem 2.2 are calculated using the true bandit means, and sampling is then performed using these fixed weights throughout.
>
> - The flat regime to the left of the dashes line in Figure 3  appears because at this point, making arm two feasible automatically makes it better than arm one. As argued in the text, the cost of the latter is \rho independent. However, the exact correlation value at which this occurs, i.e. the location of the dashed line, is instance dependent and can range from  -1 to 1. The extreme values of -1 and 1 occur when the mean reward of arm two is moved below -1/3 or above 1/3. In these cases, making arm two just feasible will always make it worse or better than arm 1. In the former case, there will not be a flat regime anywhere and in the latter case, the entire plot will be flat and neither negative nor positive correlation will seem to influence the sample complexity. In that sense, there is nothing special about negative correlations in general.
> We chose this specific instance because it clearly highlights the three regimes that might happen in terms of $ρ$: there will be a flat line, followed by two different curves, where the behaviour of the plot is determined by the different cases outlined  in Theorem 2.2 and Proposition 2.2. However, we agree that it would be useful to stress the instance dependence and to mention how the figure might change for other instances.
>
> - That is correct, and intuitively this also makes sense: the optimal algorithm that ignores correlation samples arm two a lot more frequently than arm one (as can be seen for rho=0 in the right plot of Figure 3). However, for large values of rho, the optimal algorithm (that **does** take into account correlation) will sample arm one more frequently. Therefore, for large values of rho, uniform sampling more closely resembles the optimal algorithm. This is an interesting observation that deserves to be explicitly highlighted.

---

> > ### Comment · Reviewer_MhZm · 2025-08-05
> >
> > I would like to thank the authors for their detailed and thoughtful response to my comments. I appreciate their efforts to clarify the technical aspects and to provide further intuition behind their choices. While some of my concerns have been addressed, I believe a few key points still warrant additional attention to strengthen the paper's contributions:
> >
> > - The concentration inequalities used in the proofs particularly in the Gaussian case with unknown covariance should be stated explicitly. As it stands, the derivations rely on several implicit transformations and non-trivial distributional equivalences. Making these steps transparent would significantly improve the accessibility and soundness of the arguments. To the best of my knowledge, existing analyses of GLR in the Gaussian setting with unknown variances (e.g., Jourdan et al., 2022) involve significantly more technical arguments. In that regard, Theorem 3.1 is an interesting and potentially valuable contribution in its own and would benefit from a more transparent presentation.
> > - Including an empirical comparison with the algorithm from Top Feasible Arm Identification (Katz-Samuels and Scott, 2019) would help contextualize the performance of the proposed method and clarify its practical advantages. Given that this prior work directly addresses a closely related setting, such a baseline would enhance the completeness of the experimental section.
> >
> > Despite these remaining points, I find the setting to be well-motivated and the proposed framework to offer non-trivial insights, particularly regarding the influence of correlation and the computation of oracle weights. I believe the authors could address the remaining issues in a revised version. Given the novelty in handling general models and the promising direction of this work, I now feel that the merits of the paper outweigh the weaknesses, and I will consider increasing my score accordingly.

---

> > > ### Author Response · Authors · 2025-08-06
> > >
> > > We sincerely thank the reviewer for the thoughtful follow-up and for recognising the contributions of our work.
> > >
> > > * We appreciate the comment that this result could be a valuable contribution on its own. Given its importance and nontrivial nature, we agree that the presentation deserves further clarification. In the revised version, we will elaborate on the key steps of the proof and provide additional explanation to make the argument more transparent.
> > >
> > > * We recognise the merit of including an empirical comparison of our method to that by Katz-Samuels and Scott (2019), and we will do so in the revision.
> > >
> > > We are grateful for the reviewer’s constructive feedback and believe that the planned revisions will strengthen the clarity and impact of the paper.

---

### Official Review · Reviewer_F1rN · 2025-07-01

**Clarity:** 1
**Significance:** 2
**Originality:** 2
**Rating:** 4
**Confidence:** 3

**Summary:**

The authors proposed a variant of the standard K-armed best arm identification problem, where we also have costs for the arms. The idea is to find the best arm subject to a cost constraint. Given that, every arm has 2 distributions, one for the reward and the other for the cost. The authors consider 3 scenarios: Gaussian with fixed covariance, Gaussian with unknown covariance, and non-parametric distributions with rectangular support. The authors provided lower bounds and an algorithm, which they tested compared to baselines.

**Questions:**

- Could the authors comment on the originality of the work?
- Could the authors highlight the technical challenges of the work?

**Ethical Concerns:**

["NO or VERY MINOR ethics concerns only"]

**Final Justification:**

The rebuttal solved my main concerns on the technical novelty. I still have some concerns on the originality. Given that, I increased my score from 3 to 4.

**Limitations:**

Yes. The limitations of the work are properly discussed.

**Paper Formatting Concerns:**

No formatting concerns.

**Quality:**

3

**Strengths And Weaknesses:**

### Strengths
- The work is solid from a technical perspective, and it is built on well-established BAI literature.
- The theoretical results are in line with the BAI literature, even if I haven't checked the proofs in the appendix.
- To the best of the reviewer's knowledge, related works are properly cited.
- The authors also provided an experimental campaign to validate the results.

### Weaknesses
- The main weakness of the paper is the presentation: sections 2 and 3 are hard to follow and should be revised to make them more readable to non-experts in the BAI literature. Related to the presentation, some symbols in section 2 are not formally introduced (even if it is often easy to infer them, e.g., $\gamma$).
- Another weakness of the paper is the originality: the paper presents a standard implementation of constraints in a well-known setting, and there are no characteristic/unexpected findings that the authors highlighted in the work.

---

> ### Author Rebuttal · Authors · 2025-07-31
>
> We thank the reviewer for their evaluation and for acknowledging the technical strength of the paper. Below, we address the questions and concerns raised. For clarity, reviewer comments are repeated in **_bold italics_**, followed by our responses in regular font.
>
> ## Weaknesses ##
>
> **_The main weakness of the paper is the presentation: sections 2 and 3 are hard to follow and should be revised to make them more readable to non-experts in the BAI literature. Related to the presentation, some symbols in section 2 are not formally introduced (even if it is often easy to infer them, e.g., γ)_**
>
> We agree that the presentation in Sections 2 and 3 may be somewhat terse for readers less familiar with the BAI literature. To address this, we will consider adding a brief overview of the standard approach commonly used in this setting, which our work builds upon. Additionally, we will carefully review the manuscript to ensure that all symbols and notations are clearly and properly introduced.
>
> **_Another weakness of the paper is the originality: the paper presents a standard implementation of constraints in a well-known setting, and there are no characteristic/unexpected findings that the authors highlighted in the work._**
>
> We respectfully disagree with the claim that the work lacks originality or surprising insights. We believe the contributions of the paper are novel in several ways:
> 1. The treatment of multi-dimensional rewards with arbitrary dependence remains largely unexplored in the literature (particularly in the nonparametric setting) despite being highly relevant in real-world applications. To our knowledge, we are the first to explicitly investigate this dependence structure and to demonstrate that ignoring it can lead to suboptimal performance.
> 2. We provide valid GLRT thresholds for the two-dimensional setting with dependence, which align with the lower bound on expected sample complexity. This is important because ignoring dependence leads to a different GLRT that may no longer be valid, potentially compromising the correctness of the procedure.
> 3. In terms of formulation, we incorporate feasibility constraints in a clean and general way by introducing a second cost function. This allows for a modular, plug-and-play approach to constrained best-arm identification problems, as long as cost functions are specified—something that we believe has practical and methodological value.
> 4. Beyond characterizing the instance-based complexity, we provide a generic computational recipe and an efficient algorithm given oracle access to $c\_1$ and $c\_2$.
>
> Lastly, we present (analytically for the known variance case) the effect of dependence on transportation costs, optimal sampling proportions and expected sample complexity.The interesting findings of our work is that a naive, dependence-unaware algorithm can actually perform worse than uniform allocation (Section 2.2.1 & 5). This is a counterintuitive result that highlights the need to account for dependence in multi-dimensional settings (see also our response to a question by reviewer MhZm). Moreover, we show that the oracle allocations themselves depend strongly on the correlation structure, even when all other parameters are held constant.
> We will make sure to emphasize these points more clearly in the next version.
>
> ## Questions ##
>
> **_Could the authors comment on the originality of the work?_**
>
> On top of our response to the previous point, we have attempted to position our contributions within the existing literature at the end of the introduction and to clarify our specific contributions in Section 1.1. That said, we agree that this could be made clearer. To address this, we will expand the discussion of related work in a dedicated section and more prominently highlight our own contributions.
>
> **_Could the authors highlight the technical challenges of the work?_**
>
> As noted in our response to reviewer zJnu, we believe the main challenges that make this problem worth studying are as follows:
> 1. The introduction of multi-dimensional bandits with dependencies changes the structure of the lower-bound optimisation problem, which is nontrivial even for the 2-dimensional case. In particular, there are now two ways for one arm to outperform another: by making the competing arm infeasible, or by making the original arm genuinely better. This makes it important to do a careful case-based analysis.
> 2. Beyond characterizing the instance-based complexity, we provide a generic computational recipe and an efficient algorithm given oracle access to the functions c₁ and c₂.
> 3.  In the constrained setting, it is possible that none of the arms are feasible. This differs from the standard setting and this case requires a different analysis. Handling this case with care is important, because even when a feasible arm exists in the true model, the maximum likelihood estimates may not be feasible, so we must be prepared to handle instances where no arms are feasible.
> 4. Concentration results in two dimensions are less well-developed and extending techniques that work in one dimension to the two-dimensional setting is nontrivial.

---

> > ### Comment · Reviewer_F1rN · 2025-08-08
> >
> > I would like to thanks the Authors for the response. The rebuttal solved my main concerns on the technical novelty, while I still have some concerns on the originality.
> > After having read the other reviews and rebuttals, I increased my score to 4.

---

### Official Review · Reviewer_zJnu · 2025-07-03

**Clarity:** 3
**Significance:** 3
**Originality:** 3
**Rating:** 4
**Confidence:** 4

**Summary:**

The authors consider the task of best arm identification in multi-armed-bandits in the constrained setting. Here by constrained setting the authors mean that each arm corresponds to a bi-variate random variable where the second entry of its mean must be below a threshold $\gamma$. The authors provide generic instance-specific lower bounds revealing the structure that emerges from having constraints (see Theorem 2.2.). They further instantiate their lower bounds under three arm distribution models, namely gaussian distributions with known and unknown covariance and non-parametric distributions supported on $[0,1]^2$.  Next, they propose a best arm identification algorithm that follows the track-and-stop design principle and prove it's asymptotic optimality. Finally, they complement their theoretical findings with numerical experiments.

**Questions:**

- The authors use the notion of $\delta$-correct strategy by they never defined what that is. Can you provide a definition? shouldn't this be given in the paper?
- In the case when the variance is known, is exact knowledge actually needed for the algorithm? can the authors relax their guarantees to only knowledge of say upper bounds and lower bounds of certain variance parameters?

**Ethical Concerns:**

["NO or VERY MINOR ethics concerns only"]

**Final Justification:**

I would like to maintain my score. But I consider this work to be borderline as I don't find the extension to the constrained best arm identification setting extremely challenging. There is also the fact that most of the used tools and algorithm design ideas are not very innovative and well established in prior work. For instance, I don't find Theorem 3.1. particularly new. The authors also acknowledge some limitations of their work that I believe solving would be more noteworthy, for example, going beyond known covariances or obtaining non-asymptotic guarantees in $\delta$ would be interesting.

**Limitations:**

-  The results appear to require a specific model for arms distributions. What if this information is not known?
-  The provided guarantees for the given algorithm are only asymptotic. How good are the provided algorithms when $\delta$ is not too small?

**Paper Formatting Concerns:**

None noticed

**Quality:**

3

**Strengths And Weaknesses:**

**Strengths**
- The paper is very well written making the ideas easy to grasp.
- The authors derive instance-dependent lower bounds for best arm identification in constrained multi-armed bandits. Despite the simplicity of the problem, the resulting lower bound and its proves to be different and novel in contrast with the classical unconstrained setting.
- The authors provide an algorithm whose asymptotic sample complexity matches that of the lower bound. This demonstrates the asymptotic optimality of their algorithm and tightness of their lower bound.
- The authors also consider the case where the arms are gaussian but with unknown variance which is a pleasant setting.
- Comparison with an exhaustive list of algorithms.


**Weaknesses**
-  The authors are only able to show asymptotic optimality while recent progress have been able to tackle best arm identification problems in the fixed confidence setting in the moderate regimes.
- The authors use ideas and tools for best arm identification which are by now well studied, namely, the change-of-of-measure argument for deriving lower bounds, and the TaS design principle. How challenging is it to deploy these ides in the constrained bandit setting?
- The experimental results rely on stylized thresholds for the definition of the stopping rules. This is a bit unsatisfactory, especially when the covariance is unknown.

---

> ### Author Rebuttal · Authors · 2025-07-31
>
> We thank the reviewer for the careful reading and acknowledging the theoretical and experimental contributions of our work. Below, we address the questions and concerns raised. For clarity, reviewer comments are repeated in **_bold italics_**, followed by our responses in regular font.
>
> ## Weaknesses ##
>
> **_The authors are only able to show asymptotic optimality while recent progress have been able to tackle best arm identification problems in the fixed confidence setting in the moderate regimes._**
>
> It is true that our current focus is on the high-confidence regime, which aligns with our motivation of modeling high-risk business decisions. Extending our approach to the moderate-confidence regime is an interesting and important direction for future work. That said, our goal here was to handle the added complexity introduced by constraints and two-dimensional arms in a principled way for various models. We view support for the moderate-confidence regime as complementary, and potentially addressable with adaptations of our framework.
>
> **_The authors use ideas and tools for best arm identification which are by now well studied, namely, the change-of-of-measure argument for deriving lower bounds, and the TaS design principle. How challenging is it to deploy these ideas in the constrained bandit setting?_**
>
> We would like to highlight four main difficulties that, in our view, make this problem a challenging direction of study:
> 1. The introduction of multi-dimensional bandits with dependencies changes the structure of the lower-bound optimisation problem, which is nontrivial even for the 2-dimensional case. In particular, there are now two ways for one arm to outperform another: by making the competing arm infeasible, or by making the original arm genuinely better. This makes it important to do a careful case-based analysis.
> 2. Beyond characterizing the instance-based complexity, we provide a generic computational recipe and an efficient algorithm given oracle access to the functions $c\_1$ and $c\_2$.
> 3. In the constrained setting, it is possible that none of the arms are feasible. This differs from the standard setting and this case requires a different analysis. Handling this case with care is important, because even when a feasible arm exists in the true model, the maximum likelihood estimates may not be feasible, so we must be prepared to handle instances where no arms are feasible.
> 4. Concentration results in two dimensions are less well-developed and extending techniques that work in one dimension to the two-dimensional setting is nontrivial.
>
> **_The experimental results rely on stylized thresholds for the definition of the stopping rules. This is a bit unsatisfactory, especially when the covariance is unknown._**
>
> We acknowledge that the stylized threshold is not (yet) theoretically justified generally, especially for the unknown covariance model. In practice, our simulations indicate that the stylized threshold is conservative as the error rates are below the confidence level (Section 4). Furthermore, this practice is common in the existing literature and often adopted as a pragmatic choice (Garivier and Kaufmann 2016, Optimal Best Arm Identification with Fixed Confidence, Degenne, Koolen, and Ménard 2019, Non-Asymptotic Pure Exploration by Solving Games, 2019, etc.).
>
> ## Questions ##
>
> **_The authors use the notion of δ-correct strategy by they never defined what that is. Can you provide a definition? shouldn't this be given in the paper?_**
>
> The definition is given on l.40-41, but we agree that it might deserve a more central spot.
>
> **_In the case when the variance is known, is exact knowledge actually needed for the algorithm? can the authors relax their guarantees to only knowledge of say upper bounds and lower bounds of certain variance parameters?_**
>
> In the current formulation of our known-variance model, knowledge of the full covariance matrix is indeed required. However, if domain knowledge provides partial information (such as upper or lower bounds on variance components, or known functional relationships) this could be incorporated into an alternative model. For instance, we have considered a variant in which the variances along each dimension are fixed, but the covariance between them is allowed to vary freely. In such cases, one would simply need to re-compute the relevant KLinf, $c\_1$ and $c\_2$ functions for the new model, analogous to Propositions 2.2-2.4. We conjecture that leveraging accurate domain knowledge in this way gives improved performance compared to assuming a fully unknown covariance matrix.
>
>
> ## Limitations ##
>
> **_The results appear to require a specific model for arms distributions. What if this information is not known?_**
>
> When no specific model is known for the arms, the natural approach is to assume the most general model possible. The most general setting our framework currently supports is that of distributions with a common rectangular support. This bounded model is sufficient to cover all the motivating practical cases of this paper.
>
> **_The provided guarantees for the given algorithm are only asymptotic. How good are the provided algorithms when δ is not too small?_**
>
> As noted above, our primary focus is on the high-confidence regime, which aligns with our motivation to model high-risk business decisions. Investigating the performance of the proposed algorithms in the moderate-confidence regime remains an interesting direction for future work.

---

> > ### Comment · Reviewer_zJnu · 2025-08-05
> > **Response to the Authors' Rebuttal**
> >
> > I thank the authors for their answers and clarifying some of my questions. I believe they have addressed some of my concerns and also acknowledged some of the limitations of this work. I still believe that important and challenging questions on the topic of best arm identification remain unaddressed by this work, notably a moderate regime analysis, or a solution without precise knowledge of the covariance. I think it is also important to make sure that all definitions and constants are formally presented. Other reviewers have also pointed this out.
> >
> > For now, I will maintain my evaluation of this work and make a reassessment at the end once I see how the discussion evolves with the other reviewers.

---

> > > ### Author Response · Authors · 2025-08-06
> > >
> > > We thank the reviewer for their continued engagement and are glad to hear that some concerns were resolved.
> > >
> > > Regarding the point about needing precise knowledge of the covariance, we would like to clarify that, in the Gaussian case, we consider two models: (1) the covariance matrix is fixed and shared across all arms, and (2) each arm has its own unknown covariance matrix, which may differ across arms. The second model is specifically designed to address the setting where no prior information about the covariance is available.
> > >
> > > In our initial rebuttal, we interpreted the reviewer’s earlier comment as referring to the first of these two settings, or an intermediate case between them. We will stress the difference between both models in the revised version.

---

### Decision · Program_Chairs · 2025-09-17

**Decision:**

Accept (poster)

**Comment:**

The paper considers a novel setting of best-arm identification in multi-armed bandits with vector observations (in this case a reward and a cost), where the goal is to identify the arm with the highest mean reward subject to its mean cost being below a specified threshold. The authors study the fundamental sample complexity of this problem using information-theoretic tools in detail, under the general case where the cost and reward variables can have arbitrary (correlated) joint distributions. Using a newly developed characterization of sample complexity, they then proceed to derive a workable sampling and stopping algorithm based on the standard Track-and-Stop framework.

The reviews of this paper commend the clarity of the technical exposition, the novelty in the form of the instance-dependent sample complexity characterization, and the generality of the cost-reward joint distribution modeling. On the flip side, concerns were raised during the review process about the limited novelty of the track-and-stop principle being applied, the lack of sample complexity results in the moderate confidence regime, lack of adequate comparisons to other existing related work, technical considerations surrounding concentration arguments, and clarity of certain technical details. Most of the concerns were admittedly resolved to the referees' satisfaction in the ensuing engagement with the authors, and I thank everyone for the active discussion of the paper's claims and techniques.

While the referees were unable to reach a consensus on the final recommendation, with one of them still remaining negative, I believe that the paper makes a valuable contribution to active sequential hypothesis testing in multi-armed bandits with an interesting and relevant reward-cost trade-off structure, and exposes the key considerations in learning with a clear exposition. For this reason, I recommend that the paper be accepted.